# Electrophotocatalytic hydrogenation of imines and reductive functionalization of aryl halides

Wen-Jie Kang [1], Yanbin Zhang [1] ✉, Bo Li[2] ✉ & Hao Guo [1] ✉

The open-shell catalytically active species, like radical cations or radical anions, generated by one-electron transfer of precatalysts are widely used in energy-consuming redox reactions, but their excited-state lifetimes are usually short. Here, a closed-shell thioxanthone-hydrogen anion species (**3**), which can be photochemically converted to a potent and long-lived reductant, is generated under electrochemical conditions, enabling the electrophotocatalytic hydrogenation. Notably, TfOH can regulate the redox potential of the active species in this system. In the presence of TfOH, precatalyst (**1**) reduction can occur at low potential, so that competitive $H_2$ evolution can be inhibited, thus effectively promoting the hydrogenation of imines. In the absence of TfOH, the reducing ability of the system can reach a potency even comparable to that of $Na^0$ or $Li^0$, thereby allowing the hydrogenation, borylation, stannylation and (hetero)arylation of aryl halides to construct C–H, C–B, C–Sn, and C–C bonds.

Photoredox catalysis[1–5] provides numerous opportunities for substrate activation by one-electron reduction or oxidation, which is a typical single-photon process. Consecutive photoinduced electron transfer (ConPET)[6–8], which overcomes the energetic limitation of a single visible light photon, is another efficient and useful synthetic strategy and has been widely applied in some high-energy demanding reactions like dehalogenation and further functionalization[6,7,9–17], pentafluorosulfanylation[18], carboxylation[19,20], arene oxidation[21], and Birch reduction[22] under mild conditions. Electrocatalysis harnesses the electrochemical potential to drive the reaction, thus avoiding the use of large amounts of chemical reducing[23–25] or oxidizing agents[26,27]. Photoelectrochemical reactors have been used for decades in energy and solar fuels. Photoelectrochemical reactions have been applied in organic synthesis for 40 years[28–30]. Combining the advantages of photocatalysis[31,32] and electrocatalysis[33,34], electrophotochemistry (EPC)[35] or photoelectrochemistry (PEC)[36,37] has been heavily popularized during the past few years[38] and is more and more important in organic synthesis. For example, C–H functionalization[39–50], dehalogenation functionalization[51,52], alcohol oxidation[53], C–H diamination[54], olefin difunctionalization[55,56], reductive cleavage[57,58], C–F arylation[59], and enantioselective cyanation[60,61] reactions were gradually developed

via such an electro-activated photoredox catalysis strategy. Notably, although ConPET and PEC are different in the way of generating catalytically active species, they both have the same SET process for photoexcited active species and substrates[62]. Generally, the excited-state lifetime of open-shell active species, such as radical cations[63,64] or radical anions[65,66], accessed via one-electron transfer of precatalysts is short due to the fast nonradiative decay (about picosecond timescale, Fig. 1a, b). In recent years, anionic or dianionic species have been gradually disclosed as closed-shell photocatalysts[67–70]. A long-lived closed-shell catalyst that functioned by two-electron cycling was also recently reported for oxidative transformations[71]. Because of the paired-electron configuration, they have relatively long excited-state lifetimes[68,69], which offers more possibilities for organocatalysis. Herein, we proposed a two-electron reducing electrophotocatalysis (2e⁻ EPC) strategy to generate a potent and long-lived closed-shell photoreductant by merging the versatility of photochemistry[6,7], the high chemoselectivity of electrochemistry[23–25], and the long lifetime of two-electron reduced species[68,69], which was a potential platform for broadening catalyst applications and developing new methodologies (Fig. 1c).

In our previous work[72], 9-HTXTF (**1**) was identified as a potent and long-lived photo-oxidant that oxidized *p*-xylene to provide a hydrogen

[1]Department of Chemistry, Fudan University, 2005 Songhu Road, Shanghai 200438, P.R. China. [2]Division of Chemistry and Chemical Engineering, California Institute of Technology, Pasadena, CA 91106, USA. ✉e-mail: ybzhang@nus.edu.sg; bli3@caltech.edu; Hao_Guo@fudan.edu.cn

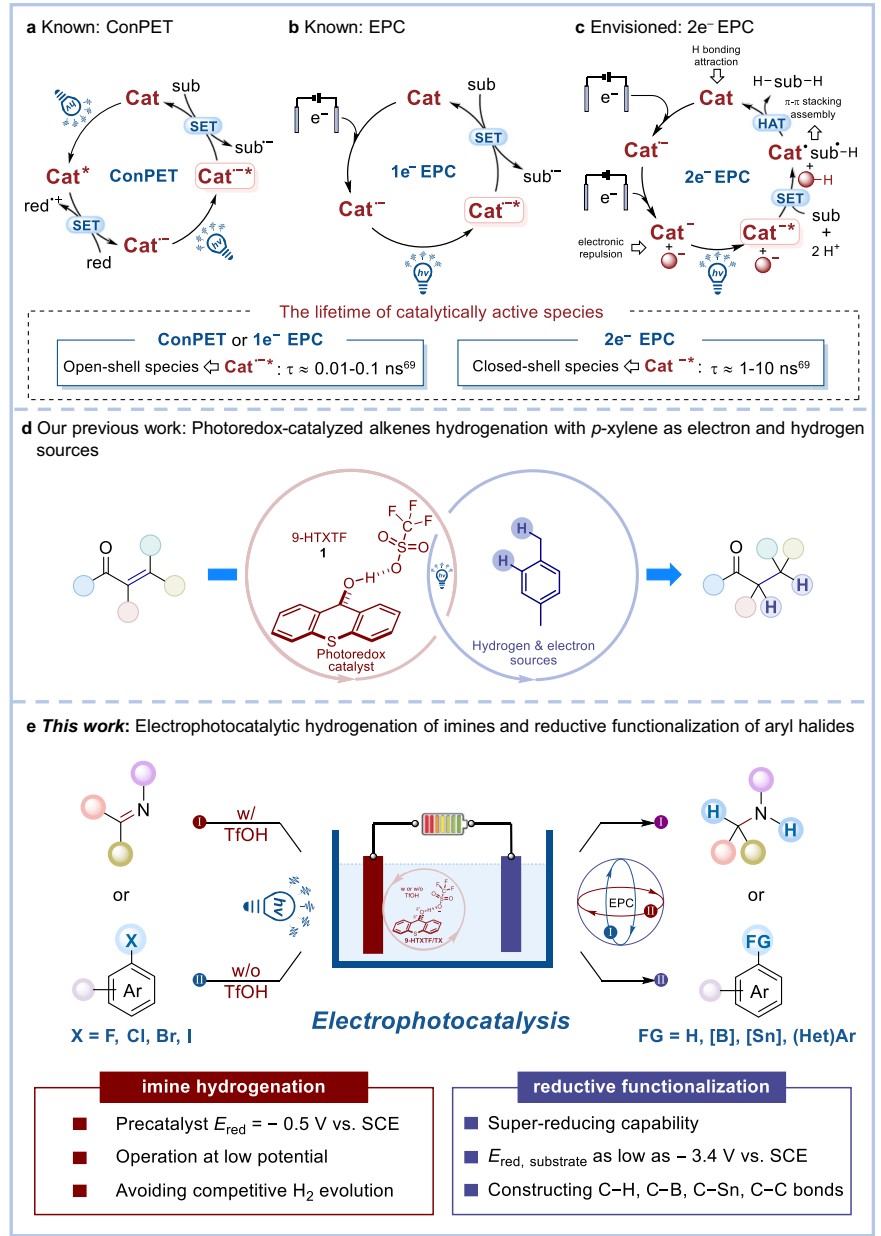

**Fig. 1 | Reaction design. a** Known ConPET strategy. **b** Known EPC strategy. **c** Envisioned 2e⁻ EPC strategy. **d** Our previous work: Photoredox-catalyzed alkenes hydrogenation with *p*-xylene as electron and hydrogen sources. **e** This work: Electrophotocatalytic hydrogenation of imines and reductive functionalization of aryl halides. Cat catalyst. red reductant. sub substrate. w/ with. w/o without.

source for alkene hydrogenation (Fig. 1d). Nevertheless, the two-electron reduced form of **1** and its electrophotocatalystic applications seem underexplored. In view of this, unveiling the reduction behavior of **1** is needed and valuable, which may help facilitate more reduction transformations. Moreover, the application of electrophotocatalytic strategy in hydrogenation reactions had been hampered due to the competitive H₂ evolution reaction. Therefore, electrophotocatalytic hydrogenation must be performed at low potential to avoid a competitive reaction. However, it is difficult to find an efficient electrophotocatalyst that can drive the hydrogenation reaction at low potential. Thereupon, in order to address the above problems, we envisioned using **1** as a new electrophotocatalyst to achieve a chemical reductant-free hydrogenation reaction at low potential. In principle, the major challenges are that (1) two-electron reduction of **1** at the cathode can smoothly generate anion species **3**, (2) this active species can be excited by visible light, (3) the resulting closed-shell molecular

has a highly negative excited-state oxidation potential to reduce substrates, (4) it possesses a long enough excited-state lifetime to collide with reactants, and (5) water can release protons (H⁺) at the anode, but it cannot be reduced to H₂ at the cathode. Notwithstanding these challenges, we report herein an efficient, transition-metal-free, and chemical reductant-free electrophotocatalytic hydrogenation of imines and reductive functionalization of aryl halides (Fig. 1e).

## Results

### Properties of the closed-shell anion species 3

Electrochemical experiments showed that a two-electron reduced state of TX (i.e., TX²⁻) could be generated by one-electron manifolds (Fig. 2b). Similarly, **1** as an analog of TX could also generate a two-electron reduced species by two-electron manifolds, forming **3** (Fig. 2b). Note that this phenomenon is in full agreement with the recently reported one-electron and two-electron manifolds of

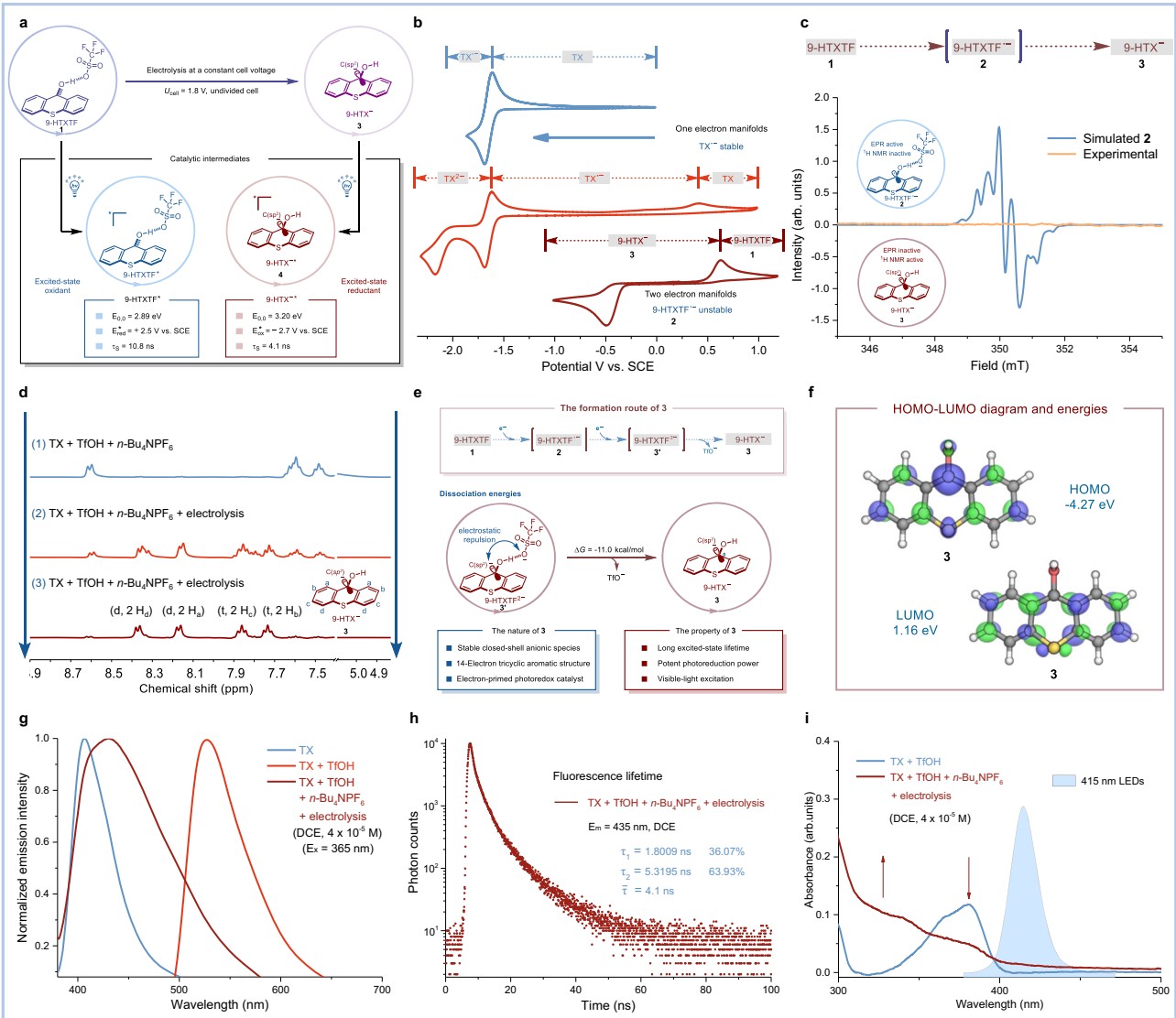

**Fig. 2 | Studies on properties of closed-shell anion species 3. a** Electricity-driven formation of **3** for electrophotocatalysis. **b** Cyclic voltammetry, **c** electron paramagnetic resonance, and **d** ¹H NMR, for details, see Supplementary Information. **e** The formation route of **3**. Free energy of dissociation was evaluated using quantum mechanical computations (see Supplementary Information for computational details). **f** HOMO−LUMO diagram and energies of **3**. **g** Fluorescence emission spectra ($\lambda_{ex}$ = 365 nm) of TX, **1** (TX (5 mM) and TfOH (10 mM)), and **3** (TX

(5 mM), TfOH (10 mM), $n$-Bu₄NPF₆ (0.2 M) and electrolysis) were collected in anhydrous DCE. **h** Fluorescence lifetime profiles ($\lambda_{em}$ = 435 nm) for **3** (TX (5 mM), TfOH (10 mM), $n$-Bu₄NPF₆ (0.2 M), and electrolysis) were collected in anhydrous DCE. **i** Absorbance profiles of **1** (TX (5 mM) and TfOH (10 mM)) and **3** (TX (5 mM), TfOH (10 mM), $n$-Bu₄NPF₆ (0.2 M) and electrolysis) were collected in anhydrous DCE. DCE dichloroethane.

nickel(II)-(pseudo)halide species[73], supporting the formation of anion species **3**. Direct evidence for whether the reduced species is open-shell radical species **2** (paramagnetic: EPR active, ¹H NMR inactive) or closed-shell anion species **3** (diamagnetic: EPR inactive, ¹H NMR active) was obtained by electron paramagnetic resonance (EPR) and ¹H NMR spectroscopic studies. The simulated EPR signal of **2** is expected to identify radical species **2** (Fig. 2c). However, no EPR signal was observed by in situ detection of the system (Fig. 2c). Notably, ¹H NMR spectroscopic studies showed the formation of a new set of peaks with the prolongation of the electrolysis time, which can be assigned to the aromatic hydrogens (Fig. 2d). Considering that no nonaromatic hydrogen signals were observed, the possibility of formation of thioxanthenol[74] (i.e., protonated **3**) could be ruled out. Combining CV, EPR, and ¹H NMR data, we reasoned that a conversion from **1** to **3** had occurred, and the reduced species was the closed-shell anionic **3** (Fig. 2e). Using quantum mechanical calculations, we found that 9-HTXTF²⁻ (**3'**) favors dissociation of the TfO⁻, likely due to the

electrostatic repulsion between **3** and TfO⁻ (Fig. 2e). Interestingly, electronic-structure calculations for **3** unveil that the highest occupied molecular orbital (HOMO) is delocalized over this 14-electron tricyclic aromatic system (Fig. 2f). The significant delocalization allows **3** to maintain a stable anionic structure and thereby prevent the formation of a C(sp³)–H bond at the 9 site. The 14-electron tricyclic aromatic system is necessary for visible light absorption. Indeed, **3** showcased a characteristic fluorescence emission peak ($\lambda_{em}$ = 435 nm, Fig. 2g) and a long excited-state lifetime ($\tau_S$ = 4.1 ns, Fig. 2h). Furthermore, the UV−vis spectrum showed that **3** could be excited by visible light (Fig. 2i). Taken together, the excited-state oxidation potential of **3** was calculated to be −2.7 V vs. SCE (Fig. 2a; For details, please see Supplementary Information), indicating that 9-HTX⁻* (**4**) possessed a sufficiently strong reductive capacity. Overall, the closed-shell anion species **3** might play a key role in electrophotocatalytic reactions due to its long excited-state lifetime, potent photoreduction power, and visible-light excitation feature.

## Table 1 | Optimization of reaction conditions[a]

Standard Conditions I
415 nm LEDs (60 W)
Al(+)/C(−), $U_{cell}$ = 1.8 V, undivided cell
TX (5 mol%), TfOH (10 mol%)
$H_2O$ (3.0 equiv.)
$n$-Bu$_4$NPF$_6$ (0.05 M)
DCE (0.01 M), Ar, 30 °C, 22 h

$E_{red}$ = −2.5 V vs. SCE

| Entry | Variation from standard conditions I | Yield[b] (%) |
|---|---|---|
| 1 | None | 87 (85)[c] |
| 2 | No photo irradiation | 0 (99) |
| 3 | No photo irradiation, 60 °C | 0 (99) |
| 4 | No photo irradiation, 80 °C | 0 (84) |
| 5 | No applied voltage | 9 (85) |
| 6 | No TX | 9 (77) |
| 7 | No TfOH | 7 (75) |
| 8 | No $H_2O$ | 13 (77) |
| 9 | No $n$-Bu$_4$NPF$_6$ | 15 (73) |
| 10 | Divided cell | 46 (46) |
| 11 | 3.0 V, 13 h | 54 (0) |
| 12 | 2.0 V, 16 h | 76 (0) |
| 13 | 1.5 V, 22 h | 74 (17) |

[a]Reaction conditions: **5** (0.2 mmol), electrode, constant voltage (U), thioxanthone (TX), TfOH, $H_2O$, electrolyte, DCE (0.01 M), undivided cell, 415 nm LEDs (60 W), 30 °C, argon atmosphere, 13–22 h.
[b]Yield and recovery were determined by $^1$H NMR analysis (400 MHz) of the crude reaction mixture using $CH_2Br_2$ (0.2 mmol) as the internal standard. Unreacted **5** in parenthesis.
[c]Isolated yield of **6**.

### Reaction development

With the understanding of the photophysical and electrochemical properties of **3**, we first examined the electrophotocatalytic hydrogenation of 5-chloro-2,3,3-trimethyl-3$H$-indole (**5**, $E_{red}$ = −2.5 V vs. SCE, Table 1) using this species. After optimization (For details, please see Supplementary Table 1), Standard Conditions I consisted of TX (5 mol%) + TfOH (10 mol%) as the catalyst, water ($H_2O$, 3.0 equiv.) as the hydrogen source, $n$-Bu$_4$NPF$_6$ (0.05 M) as the electrolyte, carbon and aluminum as cathode and anode, respectively, in anhydrous DCE with an applied cell voltage of 1.8 V (undivided cell) under visible light irradiation (415 nm LEDs) at room temperature (Table 1, entry 1). Under the above conditions, the hydrogenation product 5-chloro-2,3,3-trimethylindoline (**6**) was generated in an 85% isolated yield. Control experiments indicated that light, electricity, TX, TfOH, $H_2O$, and $n$-Bu$_4$NPF$_6$ were all necessary for reactivity (Table 1, entries 2, 5-9), which further underscored the use of electrophotocatalysis strategy. In the absence of light, increasing the reaction temperature had no effect on the product yield, which ruled out a thermochemical driving force for this reaction (Table 1, entries 2–4). When the reaction was conducted in a divided cell, the reactivity was low, possibly due to the spatial isolation of both protons generated in the anodic chamber and substrate intermediates yielded in the cathodic chamber (Table 1, entry 10). Notably, increasing the cell voltage from 1.8 V to 2.0 V (Table 1, entry 12) or 3.0 V (Table 1, entry 11) resulted in a lower yield yet a higher reaction efficiency, and decreasing the potential exhibited an acceptable yield yet a lower reactivity (Table 1, entry 13), which highlighted that the synchronous actions of photocatalytic and electrocatalytic steps were exceedingly crucial.

### Mechanistic studies

After confirming that **1** could catalyze the electrophotochemical hydrogenation of imine, we carefully studied the reaction mechanism

(Fig. 3). The test results showed that when the cell voltage was 1.8 V, the cathode potential was −0.8 V ($E_{cathode}$ = −0.8 V vs. SCE; note: The cathode potential was measured by inserting a reference electrode at the beginning of the reaction), which meant that the cathode was not able to reduce TX ($E_{red, TX}$ = −1.7 V vs. SCE) or imine **5** ($E_{red, 5}$ = −2.5 V vs. SCE), but it could reduce **1** ($E_{red, 1}$ = −0.5 V vs. SCE) into **3** ($E_{red, 5}$ < $E_{red, TX}$ < $E_{cathode}$ < $E_{red, 1}$). And, **3** could absorb the energy of a visible light photon (Fig. 2i) to form **4** that had sufficient capacity to reduce **5** ($E_{ox, 4}$ = −2.7 V vs. SCE < $E_{red, 5}$). Next, luminescence quenching experiments indicated that **5** could quench **4** with a rate constant ($k_q$) of 8.61 × $10^9$ M$^{-1}$ s$^{-1}$ according to Stern–Volmer plot (Fig. 3a, b), supporting anion species **4** for the active species that carries most of the catalytic activity. Furthermore, the NMR yields of **6** at different cathodic potentials from −0.03 to −1.23 V were collected and arranged together with the cyclic voltammogram of **1** in Fig. 3c. These data indicated that this transformation was triggered only when **1** was reduced. Light on-off experiments indicated that light is critical for imine hydrogenation, supporting the electro-activated photoredox catalysis process (Fig. 3d, ground-state **3** generated via electrocatalysis cannot drive the transformation, but photoexcited **3** (i.e., **4**) can catalyze the reaction.). Lastly, deuterium labeling experiments were performed, as shown in Fig. 3e. 3 equivalents of $D_2O$ resulted in only a partial deuterium ratio due to the trace amount of $H_2O$ in the system. As the $D_2O$ content gradually increased, the deuterium ratio of the 2 sites in **18-d** also increased accordingly. When 50 equivalents of $D_2O$ were added, the deuterium ratio of the 2 sites in **18-d** reached 92% (Fig. 3e). The above results concomitantly supported that: (1) H comes from water; (2) H is involved in the reaction as a proton. All the above results suggested that this reaction proceeded via tandem cathodic precatalyst reduction, subsequent light excitation, and following substrate reduction.

In consideration that the abovementioned redox potentials (**4** and **5**) support the conversion from **5** to **5B** in the first step (Fig. 3g), quantum mechanics computations were carried out to further probe the energetics of the second step on the basis of a model reaction (**5B → 6**, see Fig. 3f). Notably, **5B** is an electron-rich radical and exhibits a substantial Δ$G$ of 23.4 kcal/mol to be reduced by **3**. Moreover, the calculated potential ($E_{red, 5B}$ = −2.3 V vs. SCE; For details, please see Supplementary Information) excludes the possibility that **5B** is directly reduced by the cathode ($E_{cathode}$ = −0.8 V vs. SCE). One might hypothesize that **4** is capable of reducing **5B** towards carbanion **5 C**, which is indeed supported by an exergonic Δ$G$ of −22.0 kcal/mol. However, this pathway involves a bimolecular elementary process where excited-state species **4** and short-lived species **5B** need to be encountered. We propose instead that a formal intramolecular hydrogen atom transfer (HAT) occurs to transplant the H atom on the more enriched **2** to the radical site of **5B**. The calculations suggest that **2** is able to form a π-π stacking complex (**TS-1**) with **5B**. Notably, the discovery of noncovalent π-π stacking assemblies is of great importance and has been well-established in PEC reactions by Barham and co-workers[37,57]. Such a formal intramolecular HAT process avoids intermolecular collisions between two high-active and short-lived species. The accelerative effects of the nonbonded attractions allow the HAT to occur with a feasible barrier of only 11.3 kcal/mol, smoothly leading to the fully hydrogenated product **6**.

Overall, these results underscore dual modes of action for the reported electrophotocatalyst **1**, which include: (1) SET of **5 A** by the super-reducing and long-lived **4**, (2) HAT of **5B** via a π-π stacking-assisted formal intramolecular process.

On the basis of the above studies, a mechanistic rationale for this electrophotocatalytic imine hydrogenation is shown in Fig. 3g. The reaction commences with the anodic oxidation of Al into Al$^{3+}$. The latter reacts with $H_2O$ to release protons (H$^+$), meeting the demand for a hydrogen source for the electrophotocatalytic imine hydrogenation. Meanwhile, a two-electron reduced state of **1** could be generated at the

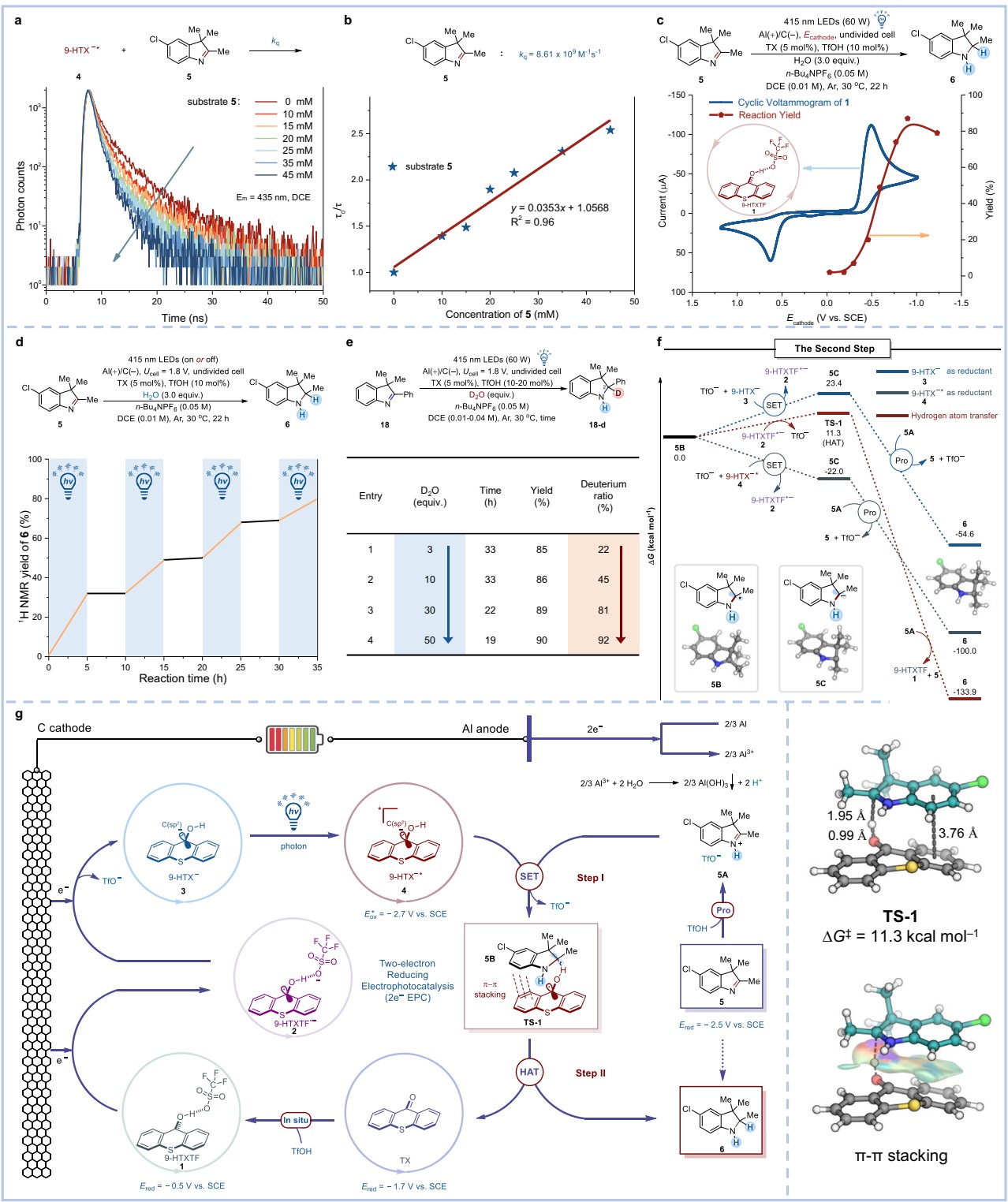

**Fig. 3 | Mechanism studies. a** Luminescence quenching experiments, **b** Stern−Volmer plot, **c** controlled potential electrolysis, **d** light on-off experiments, **e** deuterium labeling experiments, and **f** calculated free-energy profile for the electrophotocatalytic hydrogenation of **5**, for details, see Supplementary Information. **g** Proposed mechanism. SET single electron transfer. HAT hydrogen atom transfer. Pro protonation.

cathode, forming **3**. The following photoexcitation furnishes a potent reducing species **4**, which can donate an electron to protonated imine **5 A** to form a π-π stacking complex **TS-1**. Through a π-π stacking-assisted formal intramolecular HAT process, **TS-1** yields the final hydrogenated product **6** and regenerates TX.

## Substrate scope

After understanding the reaction mechanism, the scope of electrophotocatalytic imine hydrogenation was carefully explored under Standard Conditions I (Fig. 4). A variety of 3*H*-indole substrates with electronically diverse substituents could deliver the corresponding

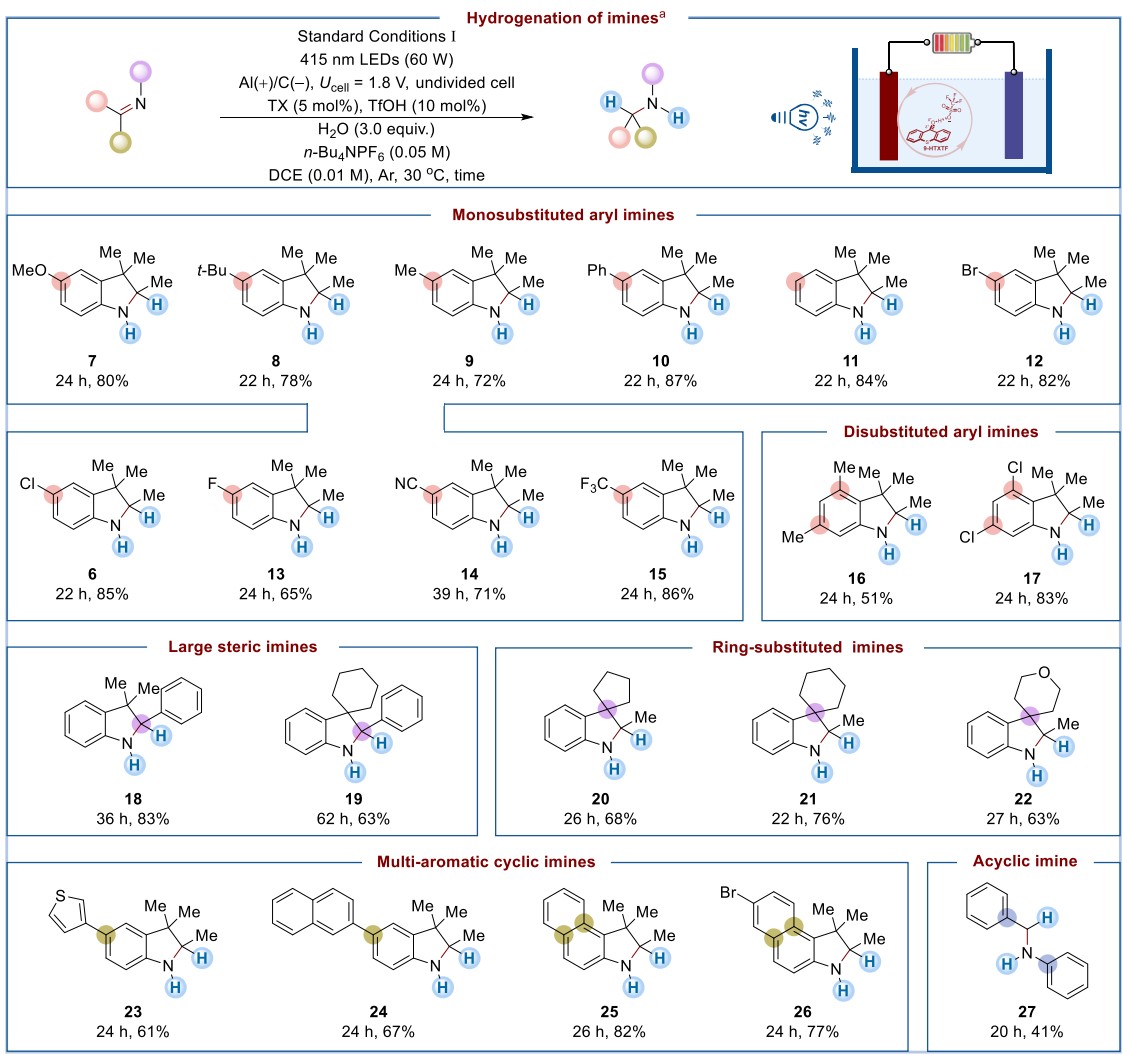

**Fig. 4 | Electrophotocatalytic hydrogenation of imines.** [a]Standard conditions I: imine substrate (0.2 mmol), TX (5 mol%), TfOH (10 mol%), $H_2O$ (3.0 equiv.), $n$-Bu$_4$NPF$_6$ (0.05 M), DCE (0.01 M), cell voltage ($U_{cell}$ = 1.8 V), Al( + )/C(−), undivided cell, 415 nm LEDs (60 W), 30 °C, argon atmosphere. Isolated yield was reported.

hydrogenation products in high yields (**7**–**11**). Potentially sensitive functional groups, such as fluorine (**13**), chlorine (**6**), bromine (**12**), nitrile (**14**), and trifluoromethyl (**15**), albeit with the apparent presence of such a strong reducing agent, were nicely tolerated, allowing the production of indolines. Given that monosubstituted aryl imines performed well in this hydrogenation reaction, disubstituted reactants with electronically differentiated substitutes were next tested (**16**–**17**). Notably, 2-aryl-substituted 3*H*-indoles (**18**–**19**), a class of substrates that were more challenging due to their inherently large steric hindrance, could also be hydrogenated under Standard Conditions I. A series of imines bearing different ring substituents at the C3 site were subjected to this electrophotocatalytic hydrogenation to obtain amine products (**20**–**22**). Multi-aromatic cyclic imines containing thiophene or naphthalene were also competent, furnishing moderate to good yields of indolines (**23-26**). Moreover, acyclic imine (**27**) also reacted but showed moderate yield, likely due to its hydrolysis under acidic conditions.

So far, we have developed the **1**-catalyzed electrophotochemical hydrogenation reaction. In order to avoid competitive H$_2$ evolution, the above reaction must be carried out at low potential. The addition of TfOH leads to the perfect precatalyst **1**, which can efficiently catalyze this reaction at low potential. But this makes the reducing power of **4** insufficient to challenge substrates with very negative

reduction potential. It can be noted from the CV that once TX is reduced, a super-reducing active species will be generated (Fig. 2b), and this species is likely to catalyze some exceptionally challenging reductions. Based on the above reasoning, TfOH was removed to further improve the reducing power of this system, which is expected to unlock the shackles of modern photoredox catalysis. After optimization, Standard Conditions II was developed for electrophotocatalytic reductive functionalization of aryl halides (Fig. 5). Reductive dehalogenation occurred without radical trapping agent (For control experiments, please see Supplementary Table 2), forming the C–H bond (**28**). In order to assess the reducing capacity of this catalytic system, some other more challenging aryl halides were used in cross-coupling reactions. For selected examples, B$_2$Pin$_2$ reacted as a coupling partner with different functional group substituted halides, including aryl chlorides (**29**–**30**), aryl bromides (**31**–**32**), and aryl iodides (**33**–**34**), to produce borylation products in moderate to excellent yields, rendering the C–B bond. To examine the scope of boronate esters, aryl chloride was chosen to couple with a variety of diboron esters, which gave the corresponding aryl borates as expected (**35**–**38**). Using hexamethylditin as a radical trapping agent, electrophotocatalytic stannylation of aryl chloride was developed, furnishing the C–Sn bond (**39**). Furthermore, aryl fluorides (**40**–**41**), chlorides (**42**–**43**), and bromides (**44**–**45**) could couple with (hetero)

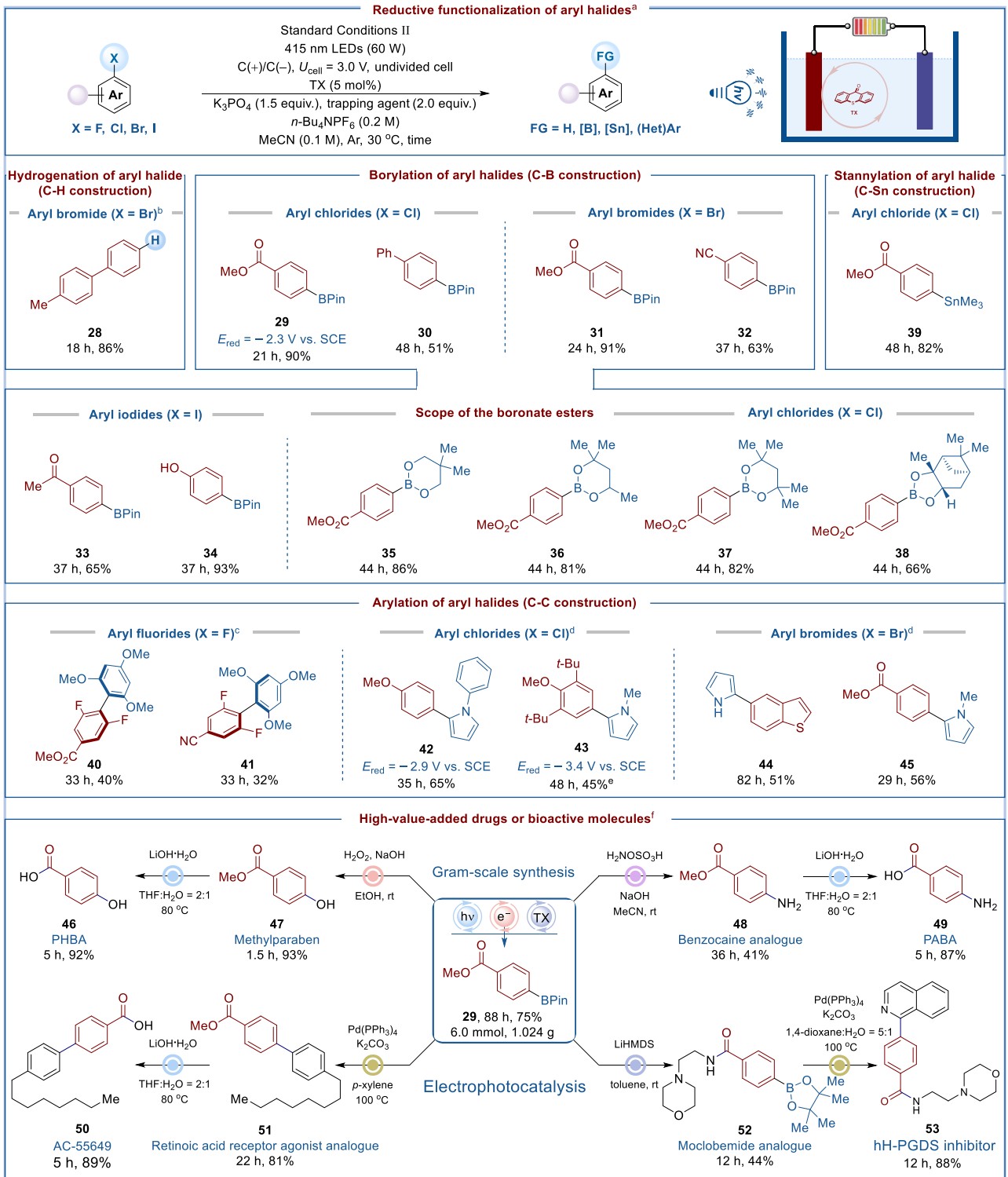

**Fig. 5 | Electrophotocatalytic reductive functionalization of aryl halides.**
[a]Standard Conditions II: aryl halide (0.4 mmol), TX (5 mol%), K₃PO₄ (1.5 equiv.), trapping agent (2.0 equiv.), $n$-Bu₄NPF₆ (0.2 M), MeCN (0.1 M), cell voltage ($U_{cell}$ = 3.0 V), C(+)/C(−), undivided cell, 415 nm LEDs (60 W), 30 °C, argon atmosphere. Isolated yield was reported. [b]No trapping agent. [c]1,3,5-Trimethoxybenzene (5.0 equiv.) was used. [d]Pyrrole (20.0 equiv.) was used. [e]K₃PO₄ (3.0 equiv.) was used. [f]For full experimental details, see Supplementary Information.

arenes in varying yields, respectively, successfully constructing the highly congested C–C bond (**40–41**). Notably, substrates **42** and **43** possess very negative reduction potentials, which are beyond the capacity of modern photoredox catalysis[51]. The running of these two exceptionally challenging substrates revealed that this electro-photocatalytic platform offered potency comparable to that of Na⁰

(−2.9 V vs. SCE) or Li⁰ (−3.3 V vs. SCE)[51]. The exergonic reduction of **43S** (precursors of compounds **43**, −3.4 V vs. SCE), which has a lower potential than **42S** (precursors of compounds **42**, −2.9 V vs. SCE), is indeed supported by the computations (For details, please see Supplementary Information). Notably, in previous reports, only systems proven to involve photoexcited radical anions could access

unactivated aryl halides[75]. Herein, such a closed-shell photoreductant could nicely achieve the above transformation. In addition, compared with previously reported electrophotocatalytic coupling reactions[51,52], our protocol has the following advantages: (1) We achieved reductive functionalization of aryl halides in an undivided cell, avoiding the need for twice as many electrolytes and solvents to use divided cells; (2) We employed radical intermediates as sacrificial agents in an undivided cell, avoiding the use of additional terminal reductants; (3) Our catalytic system could significantly improve the faradaic efficiency of electrophotocatalytic reductive functionalization (for detail, please see Supplementary Information).

## Synthetic applications

To demonstrate the practicability of this protocol, a gram-scale reaction was carried out. As shown in Fig. 5, the large-scale electrophotocatalytic reaction of aryl chloride (6.0 mmol, 1.024 g) with $B_2Pin_2$ proceeded uneventfully to provide the borylation product **29** without significant loss of yield. It is worth noting that product **29** was amenable to the synthesis of high-value-added drugs or bioactive molecules. For instance, the oxidation of **29** gave methylparaben which was an antimicrobial agent, preservative, and flavoring agent; the amination of **29** generated Benzocaine analog; the hydrolysis of **47** or **48** produced antiseptics PHBA or sunscreen PABA, respectively; the Suzuki reaction of **29** and subsequent hydrolysis yielded a highly isoform-selective agonist at the human RARβ2 receptor, AC-55649; and the amidation of **29** followed by a Suzuki reaction afforded a hH-PGDS inhibitor (**53**).

## Discussion

In conclusion, we have demonstrated that 2e[−] EPC is a feasible strategy for in situ generating a potent and long-lived closed-shell reductant **4**. Importantly, TfOH can regulate the redox potential of the catalytically active species in this system. In the presence of TfOH, this reaction can be operated at low potential. In the absence of TfOH, this system is super-reducing. Based on these findings, we develop an efficient, transition-metal-free, and chemical reductant-free electrophotocatalytic platform for hydrogenation of imines and reductive functionalization of aryl halides.

## Methods
### General procedure for electrophotocatalytic hydrogenation of imines

**Standard condition I.** An undivided cell was prepared and equipped with a stir bar. To a flame-dried 25 mL of Schlenk tube were added thioxanthone (0.01 mmol, 5 mol%), $n$-Bu$_4$NPF$_6$ (1.0 mmol, 0.05 M), anhydrous DCE (20 mL), imine substrate (0.2 mmol, 1.0 equiv.), H$_2$O (0.6 mmol, 3.0 equiv.), and TfOH (0.02 mmol, 10 mol%) under argon atmosphere. The cell was equipped with a carbon cathode and an aluminum anode and was sealed using a rubber septum and parafilm. The reaction mixture was electrolyzed at a constant cell potential of 1.8 V under irradiation of 415 nm LEDs (60 W) at 30 °C (maintained with four cooling fans). The reaction was completed as monitored by TLC (petroleum ether/ethyl acetate = 20:1). The crude product was collected by washing chamber and electrodes with EtOAc (10 mL × 3) in an ultrasonic bath. The solvent was then removed, and the residue was purified by flash chromatography on silica gel (eluent: petroleum ether/ethyl acetate = 20:1) to afford the desired product.

### General procedure for electrophotocatalytic reductive functionalization of aryl halides

**Standard condition II.** An undivided cell was prepared and equipped with a stir bar. To a flame dried 10 mL of Schlenk tube were added aryl halide (0.4 mmol, 1.0 equiv.), thioxanthone (0.02 mmol, 5 mol%), K$_3$PO$_4$ (0.6 mmol, 1.5 equiv.), trapping agent (0.8 mmol, 2.0 equiv.), $n$-Bu$_4$NPF$_6$ (0.8 mmol, 0.2 M), and anhydrous MeCN (4 mL) under argon atmosphere. The cell was equipped with a carbon cathode and a carbon anode and was sealed using a rubber septum and parafilm. The reaction mixture was electrolyzed at a constant cell potential of 3.0 V under irradiation of 415 nm LEDs (60 W) at 30 °C (maintained with four cooling fans). The reaction was completed as monitored by TLC (petroleum ether/ethyl acetate = 20:1). The crude product was collected by washing chamber, and carbon felts with EtOAc (10 mL × 3) in an ultrasonic bath. The solvent was then removed, and the residue was purified by flash chromatography on silica gel (eluent: petroleum ether/ethyl acetate = 40:1) to afford the desired product.

## Data availability
Additional experimental details, characterization, and spectra are available in the Supplementary Information. The coordinates of optimized structures are available in the Supplementary Data file. All other data are available from the corresponding author upon request.

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

## Acknowledgements
We acknowledge the Shanghai Science and Technology Committee (grant No. 21JM0010600 to H.G.) for financial support. We are grateful to Prof. Ming Gong and Prof. Haoyang Wang for their valuable discussions. We thank Xiaoya Zhao and Junyi Wang for providing fluorescence and phosphorescence spectrophotometers.

## Author contributions
W.-J.K. and H.G. conceived the project. W.-J.K. characterized anion species 3, optimized the reaction conditions, developed these reactions, evaluated the scope of these reactions, conducted the mechanism studies, and prepared the experimental portion of the manuscript and Supplementary Information. B.L. performed the (TD)-DFT calculations and prepared the computational section of the manuscript and Supplementary Information. Y.B.Z. and H.G. directed the project. All authors contributed to discussions commented on, and edited the paper.

## Competing interests
The authors declare no competing interests.
