## [Peer Review File · Nature Communications]

Reviewers' comments:

Reviewer #1 (Remarks to the Author):

Zhang, Li and Guo report that a protonated TX catalyst can be rendered a powerful reductant in a combined electro-/photocatalytic platform, for imines hydrogenation and reductive functionalization/defunctionalization aryl halides. The key synthetic advance is on the imine hydrogenation which is typically achieved of course classically very readily by metal hydride reductants or hydrogenations, but rarely by electron/proton transfer cascades in this way. Although the reaction conditions are rather technically complex for this application, conceptually neat is that the use of low applied cell potential allows authors to avoid HER reactions in their cathodic electrochemistry, achieving selective reduction of the catalyst below the HER overpotential. The authors also apply their catalytic system to aryl halide activations, including challenging aryl chloride reductions. Noteworthy was 3 eq. water can be used as the terminal reductant. (Generally) enough details are provided to reproduce experiments and manuscript is well presented.

On the conceptual and synthetic utility bases, the manuscript has potential to advance the field of interest and it may be suitable for publication in a broad interest journal such as Nat. Commun. However, mechanistic experiments at this stage are far too premature to propose 'di-anionic' species and additional evidences are needed.

#1. The authors claim a key advantage of their catalytic system is the use of a two-electron reducing EPC strategy and a potent close shell photoreductant. The proposal is disturbing in this case because:

(a) cyclic voltammogram of TX (Figure 2b) shows the second reductive wave (di-anion) is irreversible in absence of TfOH. In presence of TfOH, the first reductive wave (radical anion) is even itself unstable. This contradicts the EPR data, which show a highly persistent radical species after days of 1.8 V const. pot. electrolysis, rather than seconds of CV measurement (could be protonated radical '2').

(b) The excited state potential of -2.7 V SCE is not enough to engage aryl chloride precursors to 42 and 43 in SET.

(c) the onward reactivity of trapping aryl radicals is completely inconsistent with a photoexcited anion '4'. Aryl radical is extremely facile to reduce (~ 0.05 V SCE, JACS 2003, 125, 14801) and will certainly be immediately subsequently reduced by the co-generated ground state radical anion '2'. If authors' proposal of photoexcited anion '4' is correct, high yields of dehalogenated products would be observed. Fig. 5 shows instead high yields of aryl radical coupled products.

(d.) The EPR sample, where TX is present 20x more concentrated than reaction conditions leading to a faster reaction, took 5 days of electrolysis for the signal to decay (persists for a few days). The synthetic reaction is complete in <24 h. The decomposition of the radical anion is much slower than the time course of the reaction.

Based on ©/(d), '2' is a far stronger candidate for the active species carrying the majority of catalytic activity. Assembly of '2' with substrate (H-bonding or arene stacking) would remove need to consider lifetimes. Since the radical species is so EPR persistent, authors should irradiate the EPR active sample in the presence of substrate 5 to check for signal quenching. NMR should be taken after EPR disappears (5 days)-A closed shell species would be observable.

#2. What is their Faradaic efficiency? If the reaction is run below potentials of HER, Faradaic efficiency should be high. Authors should report the charge passed in their reactions and calculate Faradaic efficiency. Moreover, especially for constant potential electrolysis the intrinsic cell resistance varies largely between different cells/electrode batches leading to poor reproducibility; this kind of chemistry can only be reproducible when the Faradays passed are reported. Author should report this for a couple imines hydrogenations and arene functionalizations as other user needs it to reproduce

#3. Authors refer to short excited-state lifetimes of open-shell active species such as radical cations or radical anions, but no time range given for what "short" means, nor citations. No studies in Refs 35-57 reported lifetimes for excited radical cations, only one measured lifetime of an excited radical anion. Studies which give solid numbers on these lifetimes should be cited: radical anions: J. Phys. Chem. A 2000, 04, 6545; Phys. Chem. Chem. Phys. 2022, 24, 568; radical cations: J. Chem. Phys. 2023, 158, 144201; J. Am. Chem. Soc. 2018, 140, 5290

#4. - Pg1 line 32: EPC/PEC was not created in the past few years – photoelectrochemical reactors are well known for decades in energy and solar fuel applications. Photoelectrochemical processes for organic synthesis date back as far as 4 decades. They have been heavily popularized in the last few years. Authors should rephrase some points for a less misleading description of the field.

#5. ConPET is mentioned but only reductive examples are given. A seminal oxidative example has been missed: ChemCatChem 2018, 10, 2955. A few EPC early contributions have been missed: Org. Chem. Front. 2021, 8, 1132; Chem. Commun. 2021, 57, 4424

#6. Fig.3 In the literature, OER is a sluggish reaction that require forcing conditions and is usually pH sensitive. It is less represented in organic electrolytes (Inorg. Chem. 2015, 54, 11883). OER has never been reported as counter reaction in organic EPC reactions. Any evidence for this? Can authors find a supporting citation that their potentials are suitable for OER?

Minor revision:

- The naming of active 'di-anion' species is questioned, evidence of protonation is good, but it depend on how close associated the counter-anion triflate is. Also the transition orbitals of the excitation will be on the xanthone/ketyl group so it seems a photoexcited anion is more appropriate.

- Pg1 line 43: Authors refer to a "reductant-free" feature of electrochemistry. This is incorrect. electrons cannot be created for free, there are terminal reductants elsewhere in the electrochemical cell. e.g. in Ref. 23, Mg0 is sacrificed, in Ref. 22, iodide ions, etc.

- Pg1 line 34: PEC is an abbreviation for photoelectrochemistry, not “photoelectrocatalysis”. Photoelectrocatalysis refers exclusively to heterogeneous processes.
- Pg3 line 137: Authors refer to the “short-lived excited state 4” but I found no value for this compound to substantiate. Do authors mean lifetime of comparable photoex. radical anions?
- Fig. 2a: The bracket/asterisk depicting excited state is currently placed on the triflate moiety, this seems inappropriate since the transition orbitals of the excitation will be on the xanthone/ketyl group.
- Fig.3c: Authors should invert the CV so that cathodic current increases as going upward, this makes more sense to overlay with yield plot.
- Fig. 3d: I question the value of these computational results add (even if active species turns out different). ET is endergonic from a ground state and exergonic from an excited state: this is obvious? Can be put to SI file if still relevant. More useful would be computational studies on reduction potential of radical 5B or its HAT with ‘2’. Need author invoke such a complex second EPC cycle to reduce 5B, or can it be directly reduced by the cathode / undergo HAT with ‘2’ to reform TX?
- Fig. 3e: The drawn mechanism is too complex to follow, seems could be redrawn more simple without second set of electrodes on the right using compound numbers to explain transformations; e.g. 5B->5C
- Table 1: Reaction uses a 415 nm 60 W LED (radiant efficiency likely ~25%), but reaction temperature quoted is ‘rt’. These reactions cannot be 20-25 °C. So author should measure actual temp; it is known to have crucial impacts on conversions/yields in photo reactions: ChemPhotoChem 2021, 5, 808.
- Fig.5 The arrow with colored dots at the bottom of the Figure “bioactive molecules” is kind of misleading-should be removed, it suggest compounds 50/51 are transformed into 52/53 by electrophotocatalysis. I am sure this is not what author meant!
- Pg1 line 29: Authors describe electricity as “traceless”. Is this technically correct if fossil fuels are burned to make the electricity?

Reviewer #2 (Remarks to the Author):

The work submitted for publication in Nature Communications entitled “Electrophotocatalysis: hydrogenation of imines and reductive functionalization of aryl halides” by Wen-Jie Kang, Yanbin Zhang, Bo Li and Hao Guo is a new release of the past works that work better or different with electrophotocatalysis.

The SI is helpful to complete the understanding of the manuscript. There are some errors, for example in page S12 in the figure "subatrate". The same errors also are present throughout the manuscript like "posseses" at the end of the introduction (line 59).

About the discussion I would like to discuss longer with the authors of the manuscript because sentences like "which ruled out the thermochemical driving force for this reaction" are risky because it is not fully demonstrated that with 60°C it is enough to state this, and higher temperatures should be checked to keep this comment.

However, it seems that mechanistically there are 2 mechanisms, one is photocatalytic and the other thermochemical competing, but to be 100% sure here it is difficult to conclude that 60°C is enough to rule out the thermochemical.

The role of TfOH is really fascinating, and for sure makes the computational understanding more difficult. In Figure 3d I missed some computational results for the thermochemical process, in case it was feasible, or at least a proposal in the SI.

Overall, if the above "discussion" can be discussed briefly I will be pleased to accept the paper for publication.

Reviewer #3 (Remarks to the Author):

This manuscript describes a new catalyst system for electrophotocatalytic reductions, which is applied to the hydrogenation of imines and the reductive functionalization of aryl halides. Electrophotocatalysis has become an area that has experienced significant growth in recent years, and novel catalysts and approaches are of high interest. The current work utilizes thioxanthenone, which purportedly undergoes 2 electron reduction and then becomes a potent reducing photocatalyst. The yields of reactions are generally good and the scope is serviceable. As a method, this work is fine. However, there are some major issues with the presentation of the work that would prevent it from being published in its current form. Especially, the proposed mechanistic details are deeply problematic and cannot be taken seriously.

Specifically, catalyst and substrate intermediate anions are proposed that co-exist with triflic acid, an exceptionally strong proton donor. For example, structure 2 is described as a radical anion, but the protonated thioxanthenone radical is in fact a neutral species. To the extent that it may be H-bonded to the triflate anion, there must be a countercation (presumably tetrabutylammonium), but in any case, there is no interpretation in which the thioxanthenone is anionic. More problematic, structure 3 is described as a dianion. Even if the structure as drawn actually existed, it would only be a monoanion, but this is also not reasonable. Structure 3 is a carbanion, and although there would be

some amount of stabilization due to conjugation, it would still be far and away more basic than the triflate anion. Thus, 3 would surely be fully protonated in the presence of triflic acid. Furthermore, for the imine reduction, single electron reduction of 5 to 5A is not likely, because 5 is plenty basic enough to be protonated by triflic acid, and the resulting iminium ion would be much more reactive than the neutral compound. Given these simple facts, what seems most plausible is that 1 is reduced and protonated twice to form thioxanthenol, and this species serves as a potent hydride donor to the protonated imine. Photoexcitation may be operative for that step but is not obviously necessary.

The aryl halide reduction probably does proceed through the reduction / photoexcitation pathway, since triflic acid is not present in this procedure. In this sense, it is much more akin to other work in this area using anthraquinone, dicyanoanthracene, naphthalene imides, and the like. However, what is less clear is how this system differs from those established methods. Since the transformations are all basically the same as those that have been reported many times now, a comparison would be most useful.

One other issue: the introduction claims that "the short excited-state lifetime of open-shell species...remains a problem", but this statement isn't justified. Since many useful methods have been developed, a description of what problem exists and, crucially, a connection to how the current work solves that problem should be included. Otherwise, this is an unsupported statement that should be removed.

In summary, as a new methodology, this work seems perfectly fine, although it doesn't necessarily represent something groundbreaking. Mechanistically, the chemistry as presented cannot be taken seriously, due to the simple incompatibility of strongly basic intermediates and a very strong acid. The authors should reevaluate their interpretation of their results, and resubmit this work to a more specialized journal.

Point-by-point responses to reviewers' comments

Reviewer #1:

Zhang, Li and Guo report that a protonated TX catalyst can be rendered a powerful reductant in a combined electro-/photocatalytic platform, for imines hydrogenation and reductive functionalization/defunctionalization aryl halides. The key synthetic advance is on the imine hydrogenation which is typically achieved of course classically very readily by metal hydride reductants or hydrogenations, but rarely by electron/proton transfer cascades in this way. Although the reaction conditions are rather technically complex for this application, conceptually neat is that the use of low applied cell potential allows authors to avoid HER reactions in their cathodic electrochemistry, achieving selective reduction of the catalyst below the HER overpotential. The authors also applyd their catalytic system to aryl halide activations, including challenging aryl chloride reductions. Noteworthy was 3 eq. water can be used as the terminal reductant. (Generally) enough details are provided to reproduce experiments and manuscript is well presented.

On the conceptual and synthetic utility bases, the manuscript has potential to advance the field of interest and it may be suitable for publication in a broad interest journal such as Nat. Commun. However, mechanistic experiments at this stage are far too premature to propose 'di-anionic' species and additional evidences are needed.

Question 1:

#1. The authors claim a key advantage of their catalytic system is the use of a two-electron reducing EPC strategy and a potent close shell photoreductant. The proposal is disturbing in this case because:

(a) cyclic voltammogram of TX (Figure 2b) shows the second reductive wave (di-anion) is irreversible in absence of TfOH. In presence of TfOH, the first reductive wave (radical anion) is even itself unstable. This contradicts the EPR data, which show a highly persistent radical species after days of 1.8 V const. pot. electrolysis, rather than seconds of CV measurement (could be protonated radical '2').

Response:

We are very grateful to the reviewer for this suggestion. We seriously reconsidered the reason why the results of CV and EPR are contradictory to each other. We first considered whether the measured EPR signal

is the true EPR signal of **2**. Thus, we simulated the EPR signal of **2**. The results showed that the measured signal was not consistent with the simulated signal. Considering the peak shape and g value (2.003) of the measured EPR signal, and literature reports (The g values of materials with oxygen vacancies are about 2.00, which is the diagnostic data for oxygen vacancies; For references, please see: *Tractrends, Anal. Chem.* **116**, 102 (2019); *Adv. Funct. Mater.* **32**, 2109503 (2022); *Angew. Chem. Int. Ed.* **58**, 1030 (2019)), we speculated that the aluminium conductors of the carbon felts might participate in the reaction, generating oxygen vacancies, thus the corresponding EPR signal was observed (Fig. 1).

Figure 1. Electron paramagnetic resonance characterization

Accordingly, we reexamined the electrode conditions for imine hydrogenation. The results showed that the C(+)/C(-) electrodes could not make the reaction proceed, but the Al(+)/C(-) electrode could. These results indicated that aluminum (Al), not carbon (C), was the anode material that really worked during this experiment. Since the carbon felts and aluminum conductors were simultaneously immersed in the reaction solution, the reaction thus proceeded normally (Fig. 2). Many thanks to the reviewer for avoiding such a serious mistake for us. We have revised the full text to change the electrodes for imine hydrogenation from C(-)/C(+) into C(-)/Al(+).

Figure 2. Setting up the reactions

Meanwhile, this new understanding also helped us to re-understand the reaction process. As the reviewer pointed out later: at such a low potential, no oxygen evolution reaction (OER) could occur at the anode. Indeed, we speculated that the oxidation of Al is more likely to occur. The generated Al^{3+} reacts with water to release protons (H^+), thus providing hydrogen source for imine hydrogenation, which promotes the reaction to proceed smoothly (see red dotted box of Fig. 3). In this way, we also re-understood the nature for the generation of H^+ during the reaction. For more detailed evidence, please see the Response of Question 9.

Figure 3. Proposed mechanism

On the other hand, we thought that the existence of Al^{3+} affected the ^1H NMR test and thus no valid information was observed in our previous studies. In order to avoid the negative influence of aluminium conductors on the ^1H NMR and EPR tests, we re-performed the ^1H NMR and EPR experiments using C(+)/C(-) electrodes (Fig. 4).

Figure 4. Experiment equipment for ^1H NMR and EPR tests

In the ^1H NMR spectroscopic studies, a new set of peaks appeared with the extension of the electrolysis time, which could be assigned to the aromatic hydrogens. Considering that no nonaromatic hydrogen signals were observed, it can therefore be concluded that there was no formation of thioxanthenol (i.e. protonated **3**; Fig. 5). (For ^1H NMR data of thioxanthenol, please see: *Heteroatom Chemistry*, **2004**, *15*, 246-250.)

Figure 5. ^1H NMR analysis of the reduced species of **1**

In the EPR study, no signal was observed by *in situ* detection of the system (Fig. 6).

Figure 6. EPR analysis of the reduced species of **1**

Combining CV, ^1H NMR and EPR data, we reasoned that a conversion from **1** to **3** had occurred, and the reduced species was the closed-shell anionic **3**. Using quantum mechanical calculations, we found that **9-**

HTXTF²⁻ (3') favors dissociation of the TfO⁻, likely due to the electrostatic repulsion between **3** and TfO⁻ (Fig. 7).

Figure 7. The closed-shell anionic 3

Interestingly, electronic-structure calculations for **3** unveil that the highest occupied molecular orbital (HOMO) is delocalized over this 14-electron tricyclic aromatic system (Fig. 8a). The significant delocalization allows **3** to maintain a stable anionic structure and thereby prevent formation of a C(sp³)-H bond at the 9 site. The 14-electron tricyclic aromatic system is necessary for visible light absorption. Indeed, **3** showcased a characteristic fluorescence emission peak ($\lambda_{em} = 435$ nm, Fig. 8b) and a long excited-state lifetime ($\tau_s = 4.1$ ns, Fig. 8c). Furthermore, the UV-Vis spectrum showed that **3** could be excited by visible light (Fig. 8d).

Figure 8. The Photophysical properties of anionic **3**

Taken together, the excited-state oxidation potential of **3** was calculated to be -2.7 V vs. SCE (Fig. 9; For details, see Supplementary Information), indicating that **9-HTX^{-*}** (**4**) possessed a sufficiently strong reductive capacity. Overall, the anion species **3** is a stronger candidate for the active species that carries most of the catalytic activity due to its long excited-state lifetime, potent photoreduction power, and visible-light excitation feature.

Figure 9. The properties of anionic **3**

For more experimental evidence for the proof of structure **3**, please see the Response of Question 2 of the third reviewer.

Finally, we once again sincerely thank the reviewer. Thanks for helping us to avoid such serious mistakes and providing valuable suggestions.

It has been added to the revised manuscript as shown below:

“Direct evidence for whether the reduced species is open-shell radical species **2** (paramagnetic: EPR active, ^1H NMR inactive) or closed-shell anion species **3** (diamagnetic: EPR inactive, ^1H NMR active) was obtained by electron paramagnetic resonance (EPR) and ^1H NMR spectroscopic studies. The simulated EPR signal of **2** is expected to identify radical species **2** (Fig. 2c). However, no EPR signal was observed by *in situ* detection of the system (Fig. 2c). Notably, ^1H NMR spectroscopic studies showed the formation of a new set of peaks with the prolongation of the electrolysis time, which can be assigned to the aromatic hydrogens (Fig. 2d). Considering that no nonaromatic hydrogen signals were observed, the possibility of formation of thioxanthanol⁷⁰ (i.e. protonated **3**) could be ruled out. Combining CV, EPR and ^1H NMR data, we reasoned that a conversion from **1** to **3** had occurred, and the reduced species was the closed-shell anionic **3** (Fig. 2e). Using quantum mechanical calculations, we found that **9-HTXTF**²⁻ (**3**⁻) favors dissociation of the TfO⁻, likely due to the electrostatic repulsion between **3** and TfO⁻ (Fig. 2e). Interestingly, electronic-structure calculations for **3** unveil that the highest occupied molecular orbital (HOMO) is delocalized over this 14-electron tricyclic aromatic system (Fig. 2f). The significant delocalization allows **3** to maintain a stable anionic structure and thereby prevent formation of a C(sp³)–H bond at the 9 site. The 14-electron tricyclic aromatic system is necessary for visible light absorption.”

“The reaction commences with the anodic oxidation of Al into Al³⁺. The latter reacts with H₂O to release protons (H⁺), meeting the demand of hydrogen source for the electrophotocatalytic imine hydrogenation.”

Fig. 2 | Studies on properties of closed-shell anion species 3. a) Electricity-driven formation of **3** for electrophotocatalysis. b) Cyclic voltammetry, c) electron paramagnetic resonance, and d) ¹H NMR, for details, see Supplementary Information. e) The formation route of **3**. Free energy of dissociation was evaluated using quantum mechanical computations (see Supplementary Information for computational details). f) HOMO-LUMO diagram and energies of **3**. g) Fluorescence emission spectra ($\lambda_{ex} = 365$ nm) of **TX**, **1** (**TX** (5 mM) and TfOH (10 mM)), and **3** (**TX** (5 mM), TfOH (10 mM), *n*-Bu₄NPF₆ (0.2 M) and electrolysis) were collected in anhydrous DCE. h) Fluorescence lifetime profiles ($\lambda_{em} = 435$ nm) for **3** (**TX** (5 mM), TfOH (10 mM), *n*-Bu₄NPF₆ (0.2 M) and electrolysis) was collected in anhydrous DCE. i) Absorbance profiles of **1** (**TX** (5 mM) and TfOH (10 mM)) and **3** (**TX** (5 mM), TfOH (10 mM), *n*-Bu₄NPF₆ (0.2 M) and electrolysis) were collected in anhydrous DCE. ex, excitation; em, emission; DCE, dichloroethane.

The following figures have been added to the revised Supplementary Information as shown below:

Figure S7. The simulation of the EPR spectrum of 2

Spin population based on Becke method

Atomic space	Value	% of sum	% of sum abs
1(S)	0.06679737	6.679738	4.840129
2(O)	0.12295132	12.295134	8.909036
3(H)	0.00027921	0.027921	0.020231
4(C)	0.09144513	9.144515	6.626102
5(C)	-0.02897965	-2.897965	-2.099862
6(H)	-0.00008680	-0.008680	-0.006290
7(C)	0.12585307	12.585309	9.119297
8(H)	-0.00007514	-0.007514	-0.005445
9(C)	-0.03843259	-3.843259	-2.784820
10(H)	0.00011738	0.011738	0.008506
11(C)	0.11783158	11.783160	8.538060
12(H)	-0.00028986	-0.028986	-0.021003
13(C)	-0.03485511	-3.485512	-2.525597
14(C)	0.32951077	32.951083	23.876307
15(C)	-0.02973124	-2.973124	-2.154322
16(C)	0.12225720	12.225723	8.858741
17(H)	-0.00012957	-0.012957	-0.009389
18(C)	-0.03647953	-3.647953	-2.643302
19(H)	0.00008410	0.008410	0.006094

20(C)	0.12816328	12.816330	9.286694
21(H)	-0.00000586	-0.000586	-0.000425
22(C)	-0.02062560	-2.062560	-1.494528
23(H)	-0.00008926	-0.008926	-0.006468
24(C)	0.08343536	8.343537	6.045715
25(S)	0.00004508	0.004508	0.003267
26(O)	0.00118933	0.118933	0.086179
27(O)	-0.00020564	-0.020564	-0.014901
28(O)	-0.00000432	-0.000432	-0.000313
29(F)	0.00007687	0.007687	0.005570
30(F)	-0.00000956	-0.000956	-0.000693
31(F)	-0.00000059	-0.000059	-0.000042
32(C)	-0.00003691	-0.003691	-0.002675
Summing up above values:		0.99999982	
Summing up absolute value of above values:			1.38007428

Broadening

The simplest way to include line broadening is to convolute a stick spectrum with a (Gaussian or Lorentzian) lineshape after the end of the simulation. Such a convolution broadening is specified in the spin system field

lwpp.

Sys.lwpp = 0.15; % Gaussian broadening of n mT PP

B3LYP-D3(BJ)/def2-TZVP

NEVPT2/def2-TZVP

Figure S8. Electron paramagnetic resonance characterization of the reduced species

A solution of TX (0.02 mmol, 4.2 mg), TfOH (0.04 mmol, 3.5 μ L), and *n*-Bu₄NPF₆ (0.14 mmol, 54.1 mg) in CDCl₃ (3 mL) was electrolyzed at a constant cell potential of 1.8 V (C(+)/C(-)) at rt under argon atmosphere for 5 d. EPR spectrum was obtained using an EMX-8/2.7100G-18KG instrument, as shown below:

Figure S9. ^1H NMR analysis of the reduced species of 1

A solution of **TX** (0.02 mmol, 4.2 mg), TfOH (0.04 mmol, 3.5 μL), and $n\text{-Bu}_4\text{NPF}_6$ (0.14 mmol, 54.1 mg) in CDCl_3 (3 mL) was electrolyzed at a constant cell potential of 1.8 V at rt under argon atmosphere for 0-5 d.

Question 2:

(b) The excited state potential of -2.7 V SCE is not enough to engage aryl chloride precursors to 42 and 43 in SET.

Response:

Standard Conditions I: **TX** (5 mol%), TfOH (10 mol%), H_2O (3.0 equiv.), $n\text{-Bu}_4\text{NPF}_6$ (0.05 M), DCE (0.01 M), cell voltage ($U_{\text{cell}} = 1.8$ V), Al(+)/C(-), undivided cell, 415 nm LEDs (60 W), 30 $^\circ\text{C}$, argon atmosphere.

Standard Conditions II: **TX** (5 mol%), K_3PO_4 (1.5 equiv.), trapping agent (2.0 equiv.), $n\text{-Bu}_4\text{NPF}_6$ (0.2 M), MeCN (0.1 M), cell voltage ($U_{\text{cell}} = 3.0$ V), C(+)/C(-), undivided cell, 415 nm LEDs (60 W), 30 $^\circ\text{C}$, argon atmosphere.

The addition of TfOH ensured the operation of “Standard Conditions I” at low potential, which effectively inhibited the competitive HER, and thus achieved the high chemoselectivity of imine hydrogenation.

However, the reducing power of **4** is limited to -2.7 V vs. SCE. To further improve the reducing power of the catalytic system, we developed “Standard Conditions II” which removes TfOH and catalyzes the reductive functionalization reaction with **TX** alone. Here, a small point worth noting is that the precatalyst for the reductive functionalization is **TX**, and not **1** (**TX** + TfOH). The reductive functionalization of **42** and **43** fully indicate that the reducing ability of “Standard Conditions II” can reach a potency even comparable to that of Na⁰ or Li⁰.

Question 3:

(c) the onward reactivity of trapping aryl radicals is completely inconsistent with a photoexcited anion ‘4’. Aryl radical is extremely facile to reduce (~0.05 V SCE, JACS 2003, 125, 14801) and will certainly be immediately subsequently reduced by the co-generated ground state radical anion ‘2’. If authors’ proposal of photoexcited anion ‘4’ is correct, high yields of dehalogenated products would be observed. Fig. 5 shows instead high yields of aryl radical coupled products.

Response:

We achieved electrophotocatalytic imine hydrogenation using Standard Conditions I (**1** + H₂O). We used Standard Conditions II (**TX** + trapping agent) to achieve reductive functionalization of aryl halides. It is worth noting that in Standard Conditions II, we did not add hydrogen source water, but only trapping agent. Therefore, in the reductive functionalization reaction, we only obtained single aryl radical coupled products, but not dehalogenated products. As shown in the following figures:

Question 4:

(d.) The EPR sample, where TX is present 20x more concentrated than reaction conditions leading to a faster reaction, took 5 days of electrolysis for the signal to decay (persists for a few days). The synthetic reaction is complete in <24 h. The decomposition of the radical anion is much slower than the time course of the reaction.

Based on (c)/(d), '2' is a far stronger candidate for the active species carrying the majority of catalytic activity. Assembly of '2' with substrate (H-bonding or arene stacking) would remove need to consider lifetimes. Since the radical species is so EPR persistent, authors should irradiate the EPR active sample in the presence of substrate 5 to check for signal quenching. NMR should be taken after EPR disappears (5 days)-A closed shell species would be observable.

Response:

We are very grateful to the reviewer for the useful suggestions. These suggestions helped us avoid some serious mistakes. For the answer to this question, please see the Response of Question 1 for details.

In addition, the reviewer gave us very important inspirations. Two reactants can form a complex by hydrogen bonding or π - π stacking, which converts intermolecular reactions into formal intramolecular reactions, thus avoiding intermolecular collisions between two high-active and short-lived species. Theoretical calculations had shown that this strategy was feasible in our catalytic system. For more information, please see the Response of Question 16.

Question 5:

#2. What is their Faradaic efficiency? If the reaction is run below potentials of HER, Faradaic efficiency should be high. Authors should report the charge passed in their reactions and calculate Faradaic efficiency. Moreover, especially for constant potential electrolysis the intrinsic cell resistance varies largely between different cells/electrode batches leading to poor reproducibility; this kind of chemistry can only be reproducible when the Faradays passed are reported. Author should report this for a couple imines hydrogenations and arene functionalizations as other user needs it to reproduce.

Response:

We took **5** as the standard substrate for the Faradaic efficiency detection of imine hydrogenation. The charge passed was 2.2 F/mol, and the Faradaic efficiency was 83%. We took **39S** as the standard substrate for the Faradaic efficiency detection of arene functionalization. The charge passed was 1.5 F/mol, and the Faradaic efficiency was 66%.

It has been added to the Supplementary Information as shown below:

Faradaic Efficiency

1. Electrophotocatalytic imine hydrogenation

^aThe electrolysis experiments (**5**, 0.2 mmol) were conducted according to Standard Conditions I. Yield was determined by ¹H NMR analysis (400 MHz) of the crude reaction mixture using CH₂Br₂ (0.2 mmol) as the internal standard.

$$Q_1 = \frac{41.6 \text{ C}}{0.2 \text{ mmol}} = \frac{41.6}{0.2 \times 0.001 \times 96485} = 2.2 \text{ F/mol}$$

$$FE_1 = \frac{0.2 \times 0.001 \times 89\% \times 2 \times 96485}{41.6} = 83\%$$

2. Electrophotocatalytic reductive functionalization of aryl halide

^a The electrolysis experiments (**29S**, 0.4 mmol) were conducted according to Standard Conditions II. Yield was determined by ¹H NMR analysis (400 MHz) of the crude reaction mixture using CH_2Br_2 (0.2 mmol) as the internal standard.

$$Q_2 = \frac{56.4 \text{ C}}{0.4 \text{ mmol}} = \frac{56.4}{0.4 \times 0.001 \times 96485} = 1.5 \text{ F/mol}$$

$$FE_2 = \frac{0.4 \times 0.001 \times 96\% \times 96485}{56.4} = 66\%$$

Question 6:

#3. Authors refer to short excited-state lifetimes of open-shell active species such as radical cations or radical anions, but no time range given for what “short” means, nor citations. No studies in Refs 35-57 reported lifetimes for excited radical cations, only one measured lifetime of an excited radical anion. Studies which give solid numbers on these lifetimes should be cited: radical anions: J. Phys. Chem. A 2000, 04,

6545; *Phys. Chem. Chem. Phys.* 2022, 24, 568; radical cations: *J. Chem. Phys.* 2023, 158, 144201; *J. Am. Chem. Soc.* 2018, 140, 5290

Response:

It has been revised in the manuscript as shown below:

“Generally, the excited-state lifetime of open-shell active species, such as radical cations^{62,63} or radical anions^{64,65}, accessed *via* one-electron transfer of precatalysts is short due to the fast nonradiative decay (about picosecond timescale, Figs. 1a and 1b).”

62. Christensen, J. A. *et al.* Phenothiazine Radical Cation Excited States as Super-oxidants for Energy-Demanding Reactions. *J. Am. Chem. Soc.* **140**, 5290-5299 (2018).

63. Kumar, A. *et al.* Transient absorption spectroscopy based on uncompressed hollow core fiber white light proves pre-association between a radical ion photocatalyst and substrate. *J. Chem. Phys.* **158**, 144201 (2023).

64. Gosztola, D., Niemczyk, M. P., Svec, W., Lukas, A. S. & Wasielewski, M. R. Excited Doublet States of Electrochemically Generated Aromatic Imide and Diimide Radical Anions. *J. Phys. Chem. A* **104**, 6545-6551 (2000).

65. Beckwith, J. S., Aster, A. & Vauthey, E. The excited-state dynamics of the radical anions of cyanoanthracenes. *Phys. Chem. Chem. Phys.* **24**, 568-577 (2022).

Question 7:

#4. - Pg1 line 32: EPC/PEC was not created in the past few years – photoelectrochemical reactors are well known for decades in energy and solar fuel applications. Photoelectrochemical processes for organic synthesis date back as far as 4 decades. They have been heavily popularized in the last few years. Authors should rephrase some points for a less misleading description of the field.

Response:

We have revised the related description and added the corresponding citations in the manuscript as shown below:

“Photoelectrochemical reactors have been used for decades in energy and solar fuels. Photoelectrochemical reactions have been applied in organic synthesis for 40 years²⁸⁻³⁰. Combining the advantages of

photocatalysis^{31,32} and electrocatalysis^{33,34}, electrophotochemistry (EPC)³⁵ or photoelectrochemistry (PEC)^{36,37} has been heavily popularized during the past few years³⁸ and is more and more important in organic synthesis.”

28. Moutet, J.-C. & Reverdy, G. Photochemistry of cation radicals in solution : photoinduced oxidation by the phenothiazine cation radical. *Tetrahedron Lett.* **20**, 2389-2392 (1979).

29. Moutet, J.-C. & Reverdy, G. Photochemistry of cation radicals in solution; photoinduced electron-transfer reactions between alcohols and the N,N,N',N'-tetraphenyl-p-phenylenediamine cation radical. *J. Chem. Soc., Chem. Commun.*, 654-655 (1982).

30. Scheffold, R. & Orlinski, R. Synthesis and reactions of porphine-type metal complexes. 15. Carbon-carbon bond formation by light assisted B12-catalysis. Nucleophilic acylation of Michael olefins. *J. Am. Chem. Soc.* **105**, 7200-7202 (1983).

Question 8:

#5. ConPET is mentioned but only reductive examples are given. A seminal oxidative example has been missed: *ChemCatChem* 2018, 10, 2955. A few EPC early contributions have been missed: *Org. Chem. Front.* 2021, 8, 1132; *Chem. Commun.* 2021, 57, 4424

Response:

We have added the seminal oxidative example in the manuscript as shown below:

“Consecutive photoinduced electron transfer (ConPET)^{6,7}, which overcomes the energetic limitation of a single visible light photon, is another efficient and useful synthetic strategy and has been widely applied in some high-energy demanding reactions like dehalogenation and further functionalization⁶⁻¹⁶, pentafluorosulfanylation¹⁷, carboxylation^{18,19}, arene oxidation²⁰, and Birch reduction²¹ under mild conditions²².”

17. Rombach, D. & Wagenknecht, H.-A. Photoredox Catalytic Activation of Sulfur Hexafluoride for Pentafluorosulfanylation of α -Methyl- and α -Phenyl Styrene. *ChemCatChem* **10**, 2955-2961 (2018).

We have added EPC early contributions in the manuscript as shown below:

“For example, C–H functionalization³⁹⁻⁵⁰, dehalogenation functionalization^{51,52}, alcohol oxidation⁵³, C–H diamination⁵⁴, olefin difunctionalization^{55,56}, reductive cleavage^{57,58}, C–F arylation⁵⁹, and enantioselective cyanation^{60,61} reactions were gradually developed via such an electron-primed photoredox catalysis strategy.”

49. Wu, S. *et al.* Hole-mediated photoredox catalysis: tris(p-substituted)biarylammonium radical cations as tunable, precomplexing and potent photooxidants. *Org. Chem. Front.* **8**, 1132-1142 (2021).

50. Capaldo, L., Quadri, L. L., Merli, D. & Ravelli, D. Photoelectrochemical cross-dehydrogenative coupling of benzothiazoles with strong aliphatic C–H bonds. *Chem. Commun.* **57**, 4424-4427 (2021).

Question 9:

#6. Fig.3 In the literature, OER is a sluggish reaction that require forcing conditions and is usually pH sensitive. It is less represented in organic electrolytes (Inorg. Chem. 2015, 54, 11883). OER has never been reported as counter reaction in organic EPC reactions. Any evidence for this? Can authors find a supporting citation that their potentials are suitable for OER?

Response:

We thank the reviewer for helping us avoid some serious mistakes. For the answer to this question, please see the Response of Question 1 for details.

In addition, we supplemented the deuterium labeling experiments. Results indicated that 3 equivalents of D₂O resulted in only a partial deuterium ratio due to trace amount of H₂O in the system. As the D₂O content gradually increased, the deuterium ratio of the 2 site in **18-d** also increased accordingly. When 50 equivalents of D₂O was added, the deuterium ratio of the 2 site in **18-d** reached 92%. The above results concomitantly supported that: 1) H comes from water; 2) H is involved in the reaction as a proton (Table 1).

Table 1. Deuterium labeling experiments

Entry	D_2O (equiv.)	Time (h)	Yield (%) ^a	Deuterium ratio (%) ^b
1	3	33	85	22
2	10	33	86	45
3	30	22	89	81
4 ^c	50	19	90	92

And, we found that the aluminum was oxidized at the anode, and at the end of the reaction, a layer of precipitate attached to the aluminum surface.

Based on the above experimental results and phenomena, we speculate that Al is oxidized to Al^{3+} at the anode, and the resulting Al^{3+} reacts with H_2O to release protons (H^+). (Please see the red dotted box below for details).

Therefore, we revised the mechanism as follows:

Combined with the related results in our Response of Question 1, we have revised the manuscript as shown below:

“Eventually, deuterium labeling experiments were performed, as shown in Fig. 3e. 3 equivalents of D₂O resulted in only a partial deuterium ratio due to trace amount of H₂O in the system. As the D₂O content gradually increased, the deuterium ratio of the 2 site in **18-d** also increased accordingly. When 50 equivalents of D₂O was added, the deuterium ratio of the 2 site in **18-d** reached 92% (Fig. 3e). The above results concomitantly supported that: 1) H comes from water; 2) H is involved in the reaction as a proton.”

“The reaction commences with the anodic oxidation of Al into Al³⁺. The latter reacts with H₂O to release protons (H⁺), meeting the demand of hydrogen source for the electrocatalytic imine hydrogenation.”

It has been added to the Supplementary Information as shown below:

Table S4. Deuterium labeling experiments^a

Entry	TfOH (mol%)	D ₂ O (equiv.)	DCE (mL)	Time (h)	Isolated yield (%)	Deuterium ratio (%) ^b
1	10	3	20	33	85	22
2	20	10	10	33	86	45
3	20	30	5	22	89	81
4	20	50	5	19	90	92

^a Reaction conditions: **18S** (0.2 mmol), a carbon cathode and an aluminium anode, constant voltage ($U = 1.8$ V), thioxanthone (5 mol%), TfOH, D₂O, $n\text{-Bu}_4\text{NPF}_6$, anhydrous DCE, 415 nm LEDs (60 W), 30 °C, argon atmosphere, 19-33 h. ^b Deuterium ratio was determined by ¹H NMR analysis.

Question 10:

Minor revision:

- The naming of active 'di-anion' species is questioned, evidence of protonation is good, but it depend on how close associated the counter-anion triflate is. Also the transition orbitals of the excitation will be on the xanthone/ketyl group so it seems a photoexcited anion is more appropriate.

Response:

We agree that the excited transition orbitals will be on the xanthone/ketyl group, so we have changed “di-anion” into “anion species” in the manuscript as shown below:

(1) “Here, a closed-shell thioxanthone-hydrogen anion species (**9-HTX**⁻, **3**), which could be photochemically converted to a potent and long-lived reductant (E_{ox} (**4**) = - 2.7 V vs. SCE, τ_{S} (**4**) = 4.1 ns),

was generated under electrochemical conditions, enabling the electrophotocatalytic hydrogenation *via* proposed $2e^-$ EPC strategy.”

(2) “Properties of the closed-shell **anion species 3**”

(3) “Note that this phenomenon is in full agreement with the recently reported one-electron and two-electron manifolds of nickel(II)-(pseudo)halide species⁶⁹, supporting the formation of **anion species 3**.”

(4) “Direct evidence for whether the reduced species was open-shell radical species **9-HTXTF⁻** (2, paramagnetic: EPR active) or closed-shell **anion species 3** (diamagnetic: EPR inactive) was obtained by electron paramagnetic resonance (EPR) spectroscopy investigation.”

(5) “Overall, the closed-shell **anion species 3** might play a key role in electrophotocatalytic reactions due to its long excited-state lifetime, potent photoreduction power, and visible-light excitation feature⁷¹.”

Question 11:

- Pg1 line 43: Authors refer to a “reductant-free” feature of electrochemistry. This is incorrect. electrons cannot be created for free, there are terminal reductants elsewhere in the electrochemical cell. e.g. in Ref. 23, Mg0 is sacrificed, in Ref. 22, iodide ions, etc.

Response:

We have changed “reductant-free feature of electrochemistry” to “high chemoselectivity of electrochemistry” in the manuscript as shown below:

“Herein, we proposed a two-electron reducing electrophotocatalysis ($2e^-$ EPC) strategy to generate a potent and long-lived closed-shell photoreductant by merging the versatility of photochemistry^{6,7}, the **high chemoselectivity of electrochemistry**²³⁻²⁵, and the long lifetime of two-electron reduced species^{66,67}, which was a potential platform for broadening catalyst applications and developing new methodologies (Fig. 1c).”

Question 12:

- Pg1 line 34: PEC is an abbreviation for photoelectrochemistry, not “photoelectrocatalysis”. Photoelectrocatalysis refers exclusively to heterogeneous processes.

Response:

It has been revised in the manuscript as shown below:

“Combining the advantages of photocatalysis^{31,32} and electrocatalysis^{33,34}, electrophotochemistry (EPC)³⁵ or photoelectrochemistry (PEC)^{36,37} has been heavily popularized during the past few years³⁸ and is more and more important in organic synthesis.”

Question 13:

- Pg3 line 137: Authors refer to the “short-lived excited state 4” but I found no value for this compound to substantiate. Do authors mean lifetime of comparable photoex. radical anions?

Response:

The excited state lifetime of **4** is 4.1 ns (Fig. 2h).

Figure 2h. The excited state lifetime of **4**

The lifetime relationships that we described are shown below:

We have removed this description due to modification of the mechanism.

Question 14:

- Fig. 2a: The bracket/asterisk depicting excited state is currently placed on the triflate moiety, this seems inappropriate since the transition orbitals of the excitation will be on the xanthone/ketyl group.

Response:

We have revised the bracket/asterisk depicting excited state in Fig. 2a as shown below:

Question 15:

- Fig.3c: Authors should invert the CV so that cathodic current increases as going upward, this makes more sense to overlay with yield plot.

Response:

We have inverted the CV in Fig. 3c as shown below:

Question 16:

- Fig. 3d: I question the value of these computational results add (even if active species turns out different). ET is endergonic from a ground state and exergonic from an excited state: this is obvious? Can be put to SI file if still relevant. More useful would be computational studies on reduction potential of radical **5B** or its HAT with '2'. Need author invoke such a complex second EPC cycle to reduce **5B**, or can it be directly reduced by the cathode / undergo HAT with '2' to reform **TX**?

Response:

We agree with the reviewer that the reduction of **5** by photoexcited **4** is obvious and therefore provides less insights. The relevant discussion has been removed from the revised manuscript. Instead, we have performed a computational study of both the reduction potential of **5B** and the possible pathways in the second step.

We firstly evaluated the reduction potential of **5B**. To eliminate computational errors, we estimated the relative potentials between **5B** and **5**. The reduction potential of **5** had been experimentally determined to be -2.5 V vs. SCE. The calculations suggested that the reduction potential of **5B** was higher than that of **5** by 0.2 V, so we estimated that $E_{red, 5B} = -2.3$ V vs. SCE. This potential was significantly lower than the cathode potential of -0.8 V vs. SCE. Thus, we excluded the possibility that **5B** was directly reduced by the cathode.

We appreciate the reviewer for proposing the more reasonable HAT pathway, which greatly inspired us in the process of revising the manuscript. Indeed, the calculations showed that the HAT could smoothly occur in the presence of **2** to directly convert **5B** into **6** with a low barrier of only 11.3 kcal/mol. More notably, the calculations indicated that **2** is able to form a π - π stacking complex (**TS-1**) with **5B**. This entails a nonbonded attraction-driven accelerative effect that facilitates the HAT.

It has been revised in the manuscript as shown below:

“In consideration that the abovementioned redox potentials (**4** and **5**) support the conversion from **5** to **5B** in the first step (Fig. 3g), quantum mechanics computations were carried out to further probe the energetics of the second step on the basis of a model reaction (**5B** \rightarrow **6**, see Fig. 3f). Notably, **5B** is an electron-rich radical and exhibits a substantial ΔG of 23.4 kcal/mol to be reduced by **3**. Moreover, the calculated potential ($E_{\text{red, 5B}} = -2.3$ V vs. SCE; For details, please see Supplementary Information) excludes the possibility that **5B** is directly reduced by the cathode ($E_{\text{cathode}} = -0.8$ V vs. SCE). One might hypothesize that **4** is capable of reducing **5B** towards carbanion **5C**, which is indeed supported by an exergonic ΔG of -22.0 kcal/mol. However, this pathway involves a bimolecular elementary process where excited-state species **4** and short-lived species **5B** need to encounter. We propose instead that a formal intramolecular hydrogen atom transfer (HAT) occurs to transplant the H atom on the more enriched **2** to the radical site of **5B**. The calculations suggest that **2** is able to form a π - π stacking complex (**TS-1**) with **5B**. Such a formal intramolecular HAT process avoids intermolecular collisions between two high-active and short-lived species. The accelerative effects of the nonbonded attractions allow the HAT to occur with a feasible barrier of only 11.3 kcal/mol, smoothly leading to the fully hydrogenated product **6**.

Overall, these results underscore dual modes of action for the reported electrocatalyst **1**, which include: (1) SET of **5A** by the super-reducing and long-lived **4**, (2) HAT of **5B** via a π - π stacking-assisted formal intramolecular process.”

It has been revised in the Supplementary Information as shown below:

Reduction potential of **5B**

“We evaluated the reduction potential of **5B** by computational means using the same level of theory as described above. To eliminate computational errors, we estimated the relative potentials between **5B** and **5**. The reduction potential of **5** had been experimentally determined to be -2.5 V vs. SCE. The calculations

suggested that the reduction potential of **5B** was higher than that of **5** by 0.2 V, so we estimated that $E_{\text{red, 5B}} = -2.3 \text{ V vs. SCE.}$ ”

Table S5. Absolute free energies (a.u.) of calculated structures

Species	Free Energies	Species	Free Energies
1	-1935.68921	TfOH	-962.21272
2	-1935.82526	TfO ⁻	-961.80160
3	-974.12421		
3'	-1935.90830		
4	-974.05195		
5	-941.15789		
5A	-1903.40389		
5B	-941.71123		
5C	-941.77453		
6	-942.34324		
TS-1	-1915.71694		

For coordinates of optimized structures, please see Supplementary Information.

Question 17:

- Fig. 3e: The drawn mechanism is too complex to follow, seems could be redrawn more simple without second set of electrodes on the right use compound numbers to explain transformations; e.g. 5B->5C

Response:

We have redrawn the mechanism in Fig. 3g as shown below:

Question 18:

- Table 1: Reaction uses a 415 nm 60 W LED (radiant efficiency likely ~25%), but reaction temperature quoted is 'rt'. These reactions cannot be 20-25 °C. So author should measure actual temp; it is known to have crucial impacts on conversions/yields in photo reactions: ChemPhotoChem 2021, 5, 808.

Response:

The actual reaction temperature was measured to be 30 °C. And, the manuscript and supplementary information have been revised accordingly.

Typical examples are as follows:

Reductive functionalization of aryl halides^a

Question 19:

- Fig.5 The arrow with colored dots at the bottom of the Figure “bioactive molecules” is kind of misleading-should be removed, it suggest compounds 50/51 are transformed into 52/53 by electrophotocatalysis. I am sure this is no what author meant!

Response:

We completely agree and have removed the arrow as shown below:

Question 20:

- Pg1 line 29: Authors describe electricity as “traceless” . Is this technically correct if fossil fuels are burned to make the electricity?

Response:

We have revised the related description in the manuscript as shown below:

“Electrocatalysis harnesses **the electrochemical potential** to drive the reaction, thus avoiding the use of large amounts of chemical reducing²²⁻²⁴ or oxidizing agents^{25,26}.”

Reviewer #2:

The work submitted for publication in Nature Communications entitled “ Electrophotocatalysis: hydrogenation of imines and reductive functionalization of aryl halides” by Wen-Jie Kang, Yanbin Zhang, Bo Li and Hao Guo is a new release of the past works that work better or different with electrophotocatalysis.

Question 1:

The SI is helpful to complete the understanding of the manuscript. There are some errors, for example in page S12 in the figure “subatrate” . The same errors also are present throughout the manuscript like “posseses” at the end of the introduction (line 59).

Response:

Thanks to the reviewer for pointing out these mistakes. We have revised the above spelling mistakes and carefully checked the full text as shown below:

“4) it possesses a long enough excited-state lifetime to collide with reactants,”

“When the reaction was conducted in a divided cell, the reactivity was low, possibly due to the spatial isolation of both protons generated in anodic chamber and substrate intermediates yielded in cathodic chamber (Table 1, entry 10).”

Question 2:

About the discussion I would like to discuss longer with the authors of the manuscript because sentences like “which ruled out the thermochemical driving force for this reaction” are risky because it is not fully demonstrated that with 60 °C it is enough to state this, and higher temperatures should be checked to keep this comment. However, it seems that mechanistically there are 2 mechanisms, one is photocatalytic and the other thermochemical competing, but to be 100% sure here it is difficult to conclude that 60 °C is enough to rule out the thermochemical.

Response:

To sufficiently exclude the thermochemical driving force, we performed this reaction at 80 °C (the boiling point of the solvent (DCE) is 84 °C) without light (Table 1, entry 4). The results showed that increasing the reaction temperature had no effect on the product yield in the absence of light (Table 1, entries 2-4). In contrast, when the reaction was switched from dark to light at 30 °C (Table 1, entries 1-2), the reaction was initiated immediately and hydrogenated product **6** was obtained in good yield, supporting the photochemical driving force for imine hydrogenation.

Table 1. Optimization of reaction conditions^a

Entry	Variation from Standard Conditions I	Yield ^[b] (%)
1	none	87 (85) ^[c]
2	no photo irradiation	0 (99)
3	no photo irradiation, 60 °C	0 (99)
4	no photo irradiation, 80 °C	0 (84)

^a Reaction conditions: **5** (0.2 mmol), electrode, constant voltage (U), **TX**, TfOH, H₂O, electrolyte, DCE (0.01 M), undivided cell, 415 nm LEDs (60 W), 30 °C, argon atmosphere, 13-22 h. ^b Yield and recovery were determined by ¹H NMR analysis (400 MHz) of the crude reaction mixture using CH₂Br₂ (0.2 mmol) as the internal standard. Unreacted **5** in parenthesis. ^c Isolated yield of **6**.

Further, we performed Light on-off experiments. Results indicated that light is critical for imine hydrogenation, supporting the electron-primed photoredox catalysis process (Fig. 3d, ground-state **3** generated *via* electrocatalysis cannot drive the transformation, but photoexcited **3** (i.e. **4**) can catalyze the reaction.).

Figure S13. Photo irradiation on-off experiments^a

^a The electrolysis experiments were conducted according to the typical procedure III. Light was switched off during the “OFF” periods. Yield was determined by ¹H NMR analysis (400 MHz) of the crude reaction mixture using CH₂Br₂ (0.2 mmol) as the internal standard.

It has been revised in the manuscript as shown below:

“In the absence of light, increasing the reaction temperature had no effect on the product yield, which ruled out a thermochemical driving force for this reaction (Table 1, entries 2-4).”

“Light on-off experiments indicated that light is critical for imine hydrogenation, supporting the electron-primed photoredox catalysis process (Fig. 3d, ground-state **3** generated *via* electrocatalysis cannot drive the transformation, but photoexcited **3** (i.e. **4**) can catalyze the reaction.).”

Question 3:

The role of TfOH is really fascinating, and for sure makes the computational understanding more difficult. In Figure 3d I missed some computational results for the thermochemical process, in case it was feasible, or at least a proposal in the SI.

Response:

We are very grateful to the reviewer for helping us to understand the reaction mechanism more comprehensively. Control experiments and light on-off experiments indicated that light was necessary for imine hydrogenation, supporting the electron-mediated photoredox catalytic process. Further, the computations revealed that the one-electron reduction of **5** by ground-state reductant **3** requires a prohibitively high ΔG of 27.5 kcal/mol. In contrast, the photoexcited **4** allows the one-electron reduction of **3** to smoothly occur, showing an exergonic ΔG of -17.8 kcal/mol. Therefore, light is critical for the first step of the reaction. In addition, the computations suggested that the radical intermediate **5B**, once formed, will be smoothly transformed into the hydrogenated product **6** via a π - π stacking-assisted HAT. Combined with experiments and calculations, these results indicated that the initial substrate reduction step is a key reason for unfavorable thermochemical processes.

It has been added to the Supplementary Information as shown below:

“Quantum mechanical computations were carried out to probe the one-electron reduction of **5**. The calculations suggest that, in agreement with the abovementioned redox potentials (**4** and **5**), **5** undergoes an exergonic single-electron reduction by the photoexcited **9-HTX**^{-*}, showing a considerable ΔG of -17.8 kcal/mol. The ground-state **9-HTX**⁻, however, requires a prohibitively high ΔG of 27.5 kcal/mol, which explains the lack of reaction in the absence of visible light, underscoring the critical role of **9-HTX**^{-*} in initiating this energy-demanding reaction.”

Overall, if the above "discussion" can be discussed briefly I will be pleased to accept the paper for publication.

Reviewer #3:

This manuscript describes a new catalyst system for electrophotocatalytic reductions, which is applied to the hydrogenation of imines and the reductive functionalization of aryl halides. Electrophotocatalysis has become an area that has experienced significant growth in recent years, and novel catalysts and approaches are of high interest. The current work utilizes thioxanthenone, which purportedly undergoes 2 electron reduction and then becomes a potent reducing photocatalyst. The yields of reactions are generally good and the scope is serviceable. As a method, this work is fine. However, there are some major issues with the presentation of the work that would prevent it from being published in its current form. Especially, the proposed mechanistic details are deeply problematic and cannot be taken seriously.

Question 1:

Specifically, catalyst and substrate intermediate anions are proposed that co-exist with triflic acid, an exceptionally strong proton donor. For example, structure 2 is described as a radical anion, but the protonated thioxanthenone radical is in fact a neutral species. To the extent that it may be H-bonded to the triflate anion, there must be a counteraction (presumably tetrabutylammonium), but in any case, there is no interpretation in which the thioxanthenone is anionic.

Response:

We are very grateful to the reviewer for the suggestions that helped us avoid a mistake. As stated by the reviewer, the protonated thioxanthenone radical is associated with the triflate anion in the form of hydrogen bond and becomes **2**. However, the photophysical and electrochemical properties of **2** mainly depend on protonated thioxanthenone radical. Thus, it is more reasonable to describe structure **2** as a radical species.

We have changed “radical anion” to “radical species” in the manuscript as shown below:

“Direct evidence for whether the reduced species is open-shell radical species **2** (paramagnetic: EPR active, ^1H NMR inactive) or closed-shell anion species **3** (diamagnetic: EPR inactive, ^1H NMR active) was obtained by electron paramagnetic resonance (EPR) and ^1H NMR spectroscopic studies.”

“The simulated EPR signal of **2** is expected to identify radical species **2** (Fig. 2c).”

Question 2:

More problematic, structure **3** is described as a dianion. Even if the structure as drawn actually existed, it would only be a monoanion, but this is also not reasonable. Structure **3** is a carbanion, and although there would be some amount of stabilization due to conjugation, it would still be far and away more basic than the triflate anion. Thus, **3** would surely be fully protonated in the presence of triflic acid.

Question 4:

Given these simple facts, what seems most plausible is that **1** is reduced and protonated twice to form thioxanthenol, and this species serves as a potent hydride donor to the protonated imine. Photoexcitation may be operative for that step but is not obviously necessary.

Response:

Initially, we also thought that structure **3**, in the presence of TfOH, must be fully protonated to give thioxanthenol. To check this speculation, we did the following experiments:

1. Thioxanthenol was firstly synthesized according to literatures (*Chinese Chemical Letters*, **2015**, 26, 951-954; *Heteroatom Chemistry*, **2004**, 15, 246-250). However, because thioxanthenol is extremely unstable, we were only able to obtain a crude product mixture which contains thioxanthenol in 80% ¹H NMR yield. Then, we employed equivalents of thioxanthenol to react with substrate **5** under argon atmosphere in the presence of TfOH. The results showed that no product **6** was generated and the recovery of the raw material was 91%.

Synthesis and application of thioxanthenol (B)

A : B : C = 2/2 : 8.76/1 : 2.41/2

B : 80 %

2. Next, we performed simulation calculations for thioxanthanol and found that it could not be excited by visible light due to the destruction of its conjugate structure. However, our measured UV-Vis results showed that the reduced species of **1** could be excited by visible light.

Calculated UV-Vis for thioxanthanol (B)

Figure 2i. Measured UV-Vis for reduced species of 1.

3. Moreover, we repeated the control experiments and supplemented the light on-off experiments. The results indicated that light was necessary for the hydrogenation of imines. Therefore, the catalyst for this reaction must be able to absorb visible light.

Table 1. Optimization of reaction conditions^a
Entry	Variation from Standard Conditions I	Yield ^[b] (%)
1	none	87
2	no photo irradiation	0 (99)

^a Reaction conditions: **5** (0.2 mmol), electrode, constant voltage (U), **TX**, TfOH, H₂O, electrolyte, DCE (0.01 M), undivided cell, 415 nm LEDs (60 W), 30 °C, argon atmosphere, 22 h. ^b Yield and recovery were determined by ¹H NMR analysis (400 MHz) of the crude reaction mixture using CH₂Br₂ (0.2 mmol) as the internal standard. Unreacted **5** in parenthesis.

Figure S13. Photo irradiation on-off experiments^a

^aThe electrolysis experiments were conducted according to the typical procedure III. Light was switched off during the “OFF” periods. Yield was determined by ^1H NMR analysis (400 MHz) of the crude reaction mixture using CH_2Br_2 (0.2 mmol) as the internal standard.

4. Since the previous CV results suggested that **1** could produce a two-electron reduced species by two-electron manifolds (Fig. 2b). Therefore, we monitored the electroreduction process of the catalyst by EPR and ^1H NMR. In the EPR study, no signal was observed by *in situ* detection of the system (Fig. 2c). But in the ^1H NMR spectroscopic studies, a new set of peaks appeared with the extension of the electrolysis time, which could be assigned to the aromatic hydrogens (Fig. 2d). Considering that no nonaromatic hydrogen signals were observed, it can therefore be concluded that there was no formation of thioxanthanol (i.e. protonated **3**, Fig. 2d and S6).

Figure 2b-d. The reduced species of 1

For more information, please see the Response of Question 1 of Reviewer 1.

5. Taken together, we inferred that a conversion from **1** to **3** had occurred and that the reduced species was not thioxanthenol but probably **3**. Then, we performed quantum calculations for this possible structure **3**. The results show that because of the electrostatic repulsion between **3** and TfO⁻, **9-HTXTF^{2•-}** prefers to dissociate TfO⁻ (Fig. 2e).

Figure 2e. The closed-shell anionic 3

6. Interestingly, electronic-structure calculations for **3** unveil that the highest occupied molecular orbital (HOMO) is delocalized over this 14-electron tricyclic aromatic system (Fig. 2f). The significant delocalization allows **3** to maintain a stable anionic structure and thereby prevent formation of a C(sp³)-H bond at the 9 site. Importantly, the 14-electron tricyclic aromatic system is necessary for visible light absorption.

Figure 2f. HOMO-LUMO diagram and energies of the anionic 3

7. Finally, previous results suggest that **3** showcased a characteristic fluorescence emission peak ($\lambda_{em} = 435$ nm, Fig. 2g) and a long excited-state lifetime ($\tau_s = 4.1$ ns, Fig. 2h). Combining the absorbance profile, emission spectrum, and CV results, the excited-state oxidation potential of **3** was calculated to be -2.7 V vs. SCE (For details, see Supplementary Information), indicating that **9-HTX**^{*} (**4**) possessed a sufficiently strong reductive capacity. And, Stern-Volmer luminescence quenching experiments showed that substrate **5** could quench **4** (Fig. S9).

Figure 2g-h. The Photophysical properties of anionic **3**

Figure S10. Stern-Volmer luminescence quenching experiments

Take together, the anion species **3** is a stronger candidate for the active species that carries most of the catalytic activity.

We have changed “dianion” to “anion species” in the manuscript as shown below:

(1) “Here, a closed-shell thioxanthone-hydrogen anion species (**9-HTX⁻**, **3**), which could be photochemically converted to a potent and long-lived reductant (E_{ox} (**4**) = -2.7 V vs. SCE, τ_{S} (**4**) = 4.1 ns), was generated under electrochemical conditions, enabling the electrophotocatalytic hydrogenation *via* proposed $2e^-$ EPC strategy.”

(2) “Properties of the closed-shell anion species **3**”

(3) “Note that this phenomenon is in full agreement with the recently reported one-electron and two-electron manifolds of nickel(II)-(pseudo)halide species⁶⁹, supporting the formation of anion species **3**.”

(4) “Direct evidence for whether the reduced species was open-shell radical species **2** (paramagnetic: EPR active, ^1H NMR inactive) or closed-shell anion species **3** (diamagnetic: EPR inactive, ^1H NMR active) was obtained by electron paramagnetic resonance (EPR) and ^1H NMR spectroscopic studies.”

(5) “Overall, the closed-shell anion species **3** might play a key role in electrophotocatalytic reactions due to its long excited-state lifetime, potent photoreduction power, and visible-light excitation feature⁷¹.”

It has been added to the revised manuscript as shown below:

“Direct evidence for whether the reduced species is open-shell radical species **2** (paramagnetic: EPR active, ^1H NMR inactive) or closed-shell anion species **3** (diamagnetic: EPR inactive, ^1H NMR active) was obtained by electron paramagnetic resonance (EPR) and ^1H NMR spectroscopic studies. The simulated EPR signal of **2** is expected to identify radical species **2** (Fig. 2c). However, no EPR signal was observed by *in situ* detection of the system (Fig. 2c). Notably, ^1H NMR spectroscopic studies showed the formation of a new set of peaks with the prolongation of the electrolysis time, which can be assigned to the aromatic hydrogens (Fig. 2d). Considering that no nonaromatic hydrogen signals were observed, the possibility of formation of thioxanthanol⁷⁰ (i.e. protonated **3**) could be ruled out. Combining CV, EPR and ^1H NMR data, we reasoned that a conversion from **1** to **3** had occurred, and the reduced species was the closed-shell anionic **3** (Fig. 2e). Using quantum mechanical calculations, we found that **9-HTXTF**²⁻ (**3**⁻) favors dissociation of the TfO⁻, likely due to the electrostatic repulsion between **3** and TfO⁻ (Fig. 2e). Interestingly, electronic-structure calculations for **3** unveil that the highest occupied molecular orbital (HOMO) is delocalized over this 14-electron tricyclic aromatic system (Fig. 2f). The significant delocalization allows **3** to maintain a stable anionic structure and thereby prevent formation of a C(sp³)–H bond at the 9 site. The 14-electron tricyclic aromatic system is necessary for visible light absorption.”

Fig. 2 | Studies on properties of closed-shell anion species 3. a) Electricity-driven formation of **3** for electrophotocatalysis. b) Cyclic voltammetry, c) electron paramagnetic resonance, and d) ¹H NMR, for details, see Supplementary Information. e) The formation route of **3**. Free energy of dissociation was evaluated using quantum mechanical computations (see Supplementary Information for computational details). f) HOMO-LUMO diagram and energies of **3**. g) Fluorescence emission spectra ($\lambda_{ex} = 365$ nm) of TX, **1** (TX (5 mM) and TfOH (10 mM)), and **3** (TX (5 mM), TfOH (10 mM), *n*-Bu₄NPF₆ (0.2 M) and electrolysis) were collected in anhydrous DCE. h) Fluorescence lifetime profiles ($\lambda_{em} = 435$ nm) for **3** (TX (5 mM), TfOH (10 mM), *n*-Bu₄NPF₆ (0.2 M) and electrolysis) was collected in anhydrous DCE. i) Absorbance profiles of **1** (TX (5 mM) and TfOH (10 mM)) and **3** (TX (5 mM), TfOH (10 mM), *n*-Bu₄NPF₆ (0.2 M) and electrolysis) were collected in anhydrous DCE. ex, excitation; em, emission; DCE, dichloroethane.

It has been added to the revised Supplementary Information as shown below:

Figure S9. ^1H NMR analysis of the reduced species of **1**

A solution of **TX** (0.02 mmol, 4.2 mg), TfOH (0.04 mmol, 3.5 μL), and $n\text{-Bu}_4\text{NPF}_6$ (0.14 mmol, 54.1 mg) in CDCl_3 (3 mL) was electrolyzed at a constant cell potential of 1.8 V at rt under argon atmosphere for 0-5 d.

Question 3:

Furthermore, for the imine reduction, single electron reduction of **5** to **5A** is not likely, because **5** is plenty basic enough to be protonated by triflic acid, and the resulting iminium ion would be much more reactive than the neutral compound.

Response:

We have revised the mechanism based on suggestions of the reviewer. **5** was firstly protonated by TfOH to produce **5A**, followed by subsequent reduction. Please see the red box below for details.

It has been revised in the manuscript as shown below:

“The following photoexcitation furnishes a potent reducing species **4** which can donate an electron to protonated imine **5A** to form a π - π stacking complex **TS-1**.”

Question 5:

The aryl halide reduction probably does proceed through the reduction / photoexcitation pathway, since triflic acid is not present in this procedure. In this sense, it is much more akin to other work in this area using anthraquinone, dicyanoanthracene, naphthalene imides, and the like. However, what is less clear is how this system differs from those established methods. Since the transformations are all basically the same as those that have been reported many times now, a comparison would be most useful.

Response:

Electrophotochemical reductive functionalization was developed with the main purpose of comparison with the electrophotochemical imine hydrogenation. By comparing the above two classes of reactions, we propose a new strategy to regulate the redox potential of the active species in this system through TfOH. In the

presence of TfOH, precatalyst (**1**) reduction could occur at low potential, so that competitive H₂ evolution could be inhibited, thus effectively promoting the electrophotocatalytic hydrogenation of imines with high chemoselectivity. In the absence of TfOH, the reducing ability of the system could be significantly improved and reach a potency even comparable to that of Na⁰ or Li⁰, thereby allowing the hydrogenation, borylation, stannylation and (hetero)arylation of aryl halides to construct C–H, C–B, C–Sn, and C–C bonds.

In addition, compared with previously reported coupling reactions, our reductive functionalization reaction has the following advantages: (1) We achieved reductive functionalization of aryl halides in an undivided cell, avoiding the need for twice as many electrolytes and solvents using divided cells; (2) We employed radical intermediates as sacrificial agents in an undivided cell, avoiding the use of additional terminal reductants (e.g. using anthraquinone or dicyanoanthracene, Mg⁰ is sacrificed; using naphthalene imides, Et₃N (2 equiv) was added to the anodic chamber as a terminal reductant.); (3) Our catalytic system could significantly improve the faradaic efficiency of electrophotocatalytic reductive functionalization (e.g. using dicyanoanthracene, FE is 35%; in our work, FE is 66%).

Previous reports:

This work:

2. Electrophotocatalytic reductive functionalization of aryl halide

^a The electrolysis experiments (**29S**, 0.4 mmol) were conducted according to Standard Conditions II. Yield was determined by ¹H NMR analysis (400 MHz) of the crude reaction mixture using CH₂Br₂ (0.2 mmol) as the internal standard.

$$Q_2 = \frac{56.4 \text{ C}}{0.4 \text{ mmol}} = \frac{56.4}{0.4 * 0.001 * 96485} = 1.5 \text{ F/mol}$$

$$FE_2 = \frac{0.4 * 0.001 * 96\% * 96485.33}{56.4} = 66\%$$

Question 6:

One other issue: the introduction claims that "the short excited-state lifetime of open-shell species...remains a problem", but this statement isn't justified. Since many useful methods have been developed, a description of what problem exists and, crucially, a connection to how the current work solves that problem should be included. Otherwise, this is an unsupported statement that should be removed.

Response:

It has been revised in the manuscript as shown below:

“Generally, the excited-state lifetime of open-shell active species, such as radical cations^{62,63} or radical anions^{64,65}, accessed via one-electron transfer of precatalysts is short due to the fast nonradiative decay (about picosecond timescale, Figs. 1a and 1b). Very recently, a few two-electron reduced closed-shell catalysts were disclosed^{66,67}. Because of the paired-electron configuration, they have relatively long excited-state lifetimes, which offers more possibilities for organocatalysis. Herein, we proposed a two-electron reducing electrophotocatalysis ($2e^-$ EPC) strategy to generate a potent and long-lived closed-shell photoreductant by merging the versatility of photochemistry^{6,7}, the high chemoselectivity of electrochemistry²³⁻²⁵, and the long lifetime of two-electron reduced species^{66,67}, which was a potential platform for broadening catalyst applications and developing new methodologies (Fig. 1c).”

62. Christensen, J. A. *et al.* Phenothiazine Radical Cation Excited States as Super-oxidants for Energy-Demanding Reactions. *J. Am. Chem. Soc.* **140**, 5290-5299 (2018).

63. Kumar, A. *et al.* Transient absorption spectroscopy based on uncompressed hollow core fiber white light proves pre-association between a radical ion photocatalyst and substrate. *J. Chem. Phys.* **158**, 144201 (2023).

64. Gosztola, D., Niemczyk, M. P., Svec, W., Lukas, A. S. & Wasielewski, M. R. Excited Doublet States of Electrochemically Generated Aromatic Imide and Diimide Radical Anions. *J. Phys. Chem. A* **104**, 6545-6551 (2000).

65. Beckwith, J. S., Aster, A. & Vauthey, E. The excited-state dynamics of the radical anions of cyanoanthracenes. *Phys. Chem. Chem. Phys.* **24**, 568-577 (2022).

In summary, as a new methodology, this work seems perfectly fine, although it doesn't necessarily represent something groundbreaking. Mechanistically, the chemistry as presented cannot be taken seriously, due to the

simple incompatibility of strongly basic intermediates and a very strong acid. The authors should reevaluate their interpretation of their results, and resubmit this work to a more specialized journal.

Zhang, Li and Guo's revised manuscript did a great job at meeting previous concerns over the mechanism. Particularly, I was pleased to see the authors' due diligence in testing the control reaction with aluminium conductor/support alone instead of with the carbon felt loaded on, which crucially revealed that the carbon felt was not even necessary. I am very glad to have flagged the inconsistency with between EPR and CV data in the first place, and even more glad the authors' took this comment seriously and avoided making a mistake (convincing match between the measured g value and literature support that this EPR signal is instead coming from oxygen vacancies). The computational studies showing exergonic dissociation of triflate to give the 9-HTX anion as the active species was also convincing. Control reactions such as i) measuring actual reaction temperature; ii) a reaction at 80 oC to convincingly rule out a thermochemical process, and iii) use of thioxanthanol as a hydride donor which failed to reduce the imine substrate, were appreciated. The superposition of increasing reaction yield with the voltammogram of 5 seems to support an electron transfer mechanism.

I am now convinced by the proposed mechanism and would be open to support publication in Nat. Commun. after minor revisions below:

- Authors now propose a non-covalent pi stacking assembly (TS-1) in their photoelectrochemical reactions. It should be mentioned that the discovery and critical importance of non-covalent pi stacking assemblies in PEC reactions are well-established in the field by Barham and co-workers (Refs 47, 57), which were also calculated with slightly endergonic - but accessible - energies with respect to the components.

- The demonstration of 9-TX anion as the actual photocatalyst was convincing (excited state potential -2.7 V vs SCE), however, something that was not explained is the ability to reduce precursors of compounds 42 and 43 (-2.9 V and -3.4 V vs SCE), these are endergonic by 200-700 mV(!). For photoinduced electron transfer to be feasible for a relatively short (albeit not ultrashort) ~4 ns-lived excited state, the difference in potentials must be neutral (zero) or exergonic. Therefore, very likely a pi-stacked assembly between 9-TX anion and aryl chlorides also occurs, that assists electron transfer and simultaneously assists C(sp²)-X cleavage.

- The authors are pushing a concept of "2e EPC". However, this is conceptually misleading readers into thinking that the each catalytic cycle involves a two electron transformation of the catalyst. In fact, it is still a 1 electron EPC mechanism switching between an anion and a radical. This conceptual part of the introduction needs to be redrawn in a way that actually represents the discovery.

- An analogous concept of consecutive PET is mentioned, and one review (Ref 22) is cited. However, this citation should come earlier upon the first mention (together with references 6,7). The existence of this concept is simply stated but not elaborated; no comparisons with the photoelectrochemistry – the main topic of this paper – are given (at least, the comparison is not clear to my eyes). A review should be added that compares the concepts: *Beilstein J. Org. Chem.* 2023, 19, 1055.

- Since authors discovered the reaction involved a photoexcited anion, a key review on photoexcited anions in organic transformations should really be cited: *Angew. Chem. Int. Ed.* 2020, 60, 6270. A recent paper from Melchiorre group (*Angew. Chem. Int. Ed.* 2023, 62, e202306364) that involves a photoexcited anion for unactivated aryl halide cleavages should be cited.

- The statement "Very recently, a few two-electron reduced closed-shell catalysts were disclosed" should be revisited/expanded upon to not mislead readers. Firstly, in both these reports, the reductant was prepared by chemical reduction (sodium bisulfite or NaBH₄), not electrochemical reduction. Although for Wenger's report the phrase 'two-electron reduction' would be correct, a hydride reduction (Nocera report) is not the same as a two-electron reduction. Authors could instead mention 'anionic or dianionic catalysts'. Secondly, the 9-TX anion herein achieved activation of even deactivated

aryl halides (42 and 43), while Refs 66/67 were limited to activated aryl halides. Authors should mention that only systems proven to involve photoexcited radical anions could access unactivated aryl halides (ACS Catal. 2023, 13, 9392), and yet their system under review (9-TX anion) is a closed-shell photoreductant that can do so.

- In the pi-stacking complex TS-1, arene centroid distance should be shown. A distance of 4-6 Angstroms would support the notion of this interaction.

- Authors should mention that a closed-shell catalyst with long lifetime that functions by two-electron cycling was recently reported for oxidative transformations: Angew. Chem. Int. Ed. 2023, e202307550. Exactly analogously, i) a benzophenone-type precatalyst is reduced to give the active photocatalyst and ii) presence of a strong acid caused a profound change in redox potential for cathodic reduction.

- (Regarding the response to another reviewer on the comparison of aryl halide reductions) I would encourage authors to highlight the beneficial advantages of their protocol within the manuscript itself, (undivided cell, Faradaic efficiency, no sacrificial reductants etc) as these could greatly benefit readers when selecting reaction conditions. The ability to do these reactions in a simple undivided cell without sacrificial amine additives is indeed attractive.

- pg2, line 39. The authors should rephrase 'electron-primed' to 'electro-activated' as a more general term. Since, in many of the papers cited these are in an oxidative direction so the catalyst is not primed by an electron (rather, by a hole).

- Fig 5. The different colored circular/button symbols under the word 'electrophotocatalysis' all link to non-photoelectrochemical processes elsewhere in the scheme. Therefore it doesn't make sense to me / is misleading to have these colored icons under the word 'electrophotocatalysis'.

- Since the emission of TX depends heavily on the reaction conditions (Figure 2g). the detection wavelength for the TCSPC analysis (Figure 3a) should be given as it is in Figure 2h.

Reviewer #2 (Remarks to the Author):

The first revision of the work submitted for publication in Nature Communications entitled "Electrophotocatalysis: hydrogenation of imines and reductive functionalization of aryl halides" by Wen-Jie Kang, Yanbin Zhang, Bo Li and Hao Guo has answered in great detail all my demands and those of the other reviewers. Formatting errors aside, the discussion of how it affects temperature and/or light is simply very enriching. The triflate discussion is also appreciated. And the rest of discussion is also very fascinating. Thus, again I recommend publication.

Reviewer #3 (Remarks to the Author):

The authors have made substantial corrections to the manuscript, removing some of the errors depicting intermediates. They have also provided evidence that neutral thioxanthenol itself is not active as a photoreductant. What seems clear is that the thioxanthenone is being reduced to a species that then reduces the imine substrates, and that the process is electrophotocatalytic.

I still find the invocation of a carbanion in the presence of more acidic bonds problematic. In particular, I don't see how structure 3 would exist under these, or really any, conditions. Why wouldn't the proton on the hydroxyl group migrate to the carbon? To put it another way, imagine taking thioxanthenol and treating it with 1 equiv of strong base. Would 3 form, or would the corresponding alkoxide? I think it's the latter. On the other hand, it seems possible that the oxygen is not protonated but rather coordinated to the aluminum ions that are invoked in the mechanism. Frankly, I don't know how to easily answer this question, but if the authors want to draw the carbanion as with structure 3, I suppose they can do so.

In summary, I think that there is interesting reactivity here that is worth publishing. The mechanistic proposal is controversial in some respects, but seems like it must be accurate in its broad-stroke analysis. I support accepting this work.

Point-by-point responses to reviewers' comments

Reviewer #1:

Zhang, Li and Guo's revised manuscript did a great job at meeting previous concerns over the mechanism. Particularly, I was pleased to see the authors' due diligence in testing the control reaction with aluminium conductor/support alone instead of with the carbon felt loaded on, which crucially revealed that the carbon felt was not even necessary. I am very glad to have flagged the inconsistency with between EPR and CV data in the first place, and even more glad the authors' took this comment seriously and avoided making a mistake (convincing match between the measured g value and literature support that this EPR signal is instead coming from oxygen vacancies). The computational studies showing exergonic dissociation of triflate to give the 9-HTX anion as the active species was also convincing. Control reactions such as i) measuring actual reaction temperature; ii) a reaction at 80 oC to convincingly rule out a thermochemical process, and iii) use of thioxanthanol as a hydride donor which failed to reduce the imine substrate, were appreciated. The superposition of increasing reaction yield with the voltammogram of 5 seems to support an electron transfer mechanism.

I am now convinced by the proposed mechanism and would be open to support publication in Nat. Commun. after minor revisions below:

Question 1:

- Authors now propose a non-covalent pi stacking assembly (TS-1) in their photoelectrochemical reactions. It should be mentioned that the discovery and critical importance of non-covalent pi stacking assemblies in PEC reactions are well-established in the field by Barham and co-workers (Refs 47, 57), which were also calculated with slightly endergonic - but accessible - energies with respect to the components.

Response:

It has been added to the revised manuscript as shown below:

“Notably, the discovery of non-covalent π - π stacking assemblies is of great importance and has been well-established in PEC reactions by Barham and co-workers^{37,57}.”

37. Wu, S., Kaur, J., Karl, T. A., Tian, X. & Barham, J. P. Synthetic Molecular Photoelectrochemistry: New Frontiers in Synthetic Applications, Mechanistic Insights and Scalability. *Angew. Chem. Int. Ed.* 61, e202107811 (2022).

57. Tian, X. *et al.* Electro-mediated PhotoRedox Catalysis for Selective C(sp³)-O Cleavages of Phosphinated Alcohols to Carbanions. *Angew. Chem. Int. Ed.* 60, 20817-20825 (2021).

Question 2:

- The demonstration of 9-TX anion as the actual photocatalyst was convincing (excited state potential -2.7 V vs SCE), however, something that was not explained is the ability to reduce precursors of compounds 42 and 43 (-2.9 V and -3.4 V vs SCE), these are endergonic by 200-700 mV(!). For photoinduced electron transfer to be feasible for a relatively short (albeit not ultrashort) ~4 ns-lived excited state, the difference in potentials must be neutral (zero) or exergonic. Therefore, very likely a pi-stacked assembly between 9-TX anion and aryl chlorides also occurs, that assists electron transfer and simultaneously assists C(sp²)-X cleavage.

Response:

Standard Conditions I: TX (5 mol%), TfOH (10 mol%), H₂O (3.0 equiv.), *n*-Bu₄NPF₆ (0.05 M), DCE (0.01 M), cell voltage ($U_{\text{cell}} = 1.8$ V), Al(+)/C(-), undivided cell, 415 nm LEDs (60 W), 30 °C, argon atmosphere.

Standard Conditions II: TX (5 mol%), K₃PO₄ (1.5 equiv.), trapping agent (2.0 equiv.), *n*-Bu₄NPF₆ (0.2 M), MeCN (0.1 M), cell voltage ($U_{\text{cell}} = 3.0$ V), C(+)/C(-), undivided cell, 415 nm LEDs (60 W), 30 °C, argon atmosphere.

Catalytically active species

highly active and unstable

The addition of TfOH ensured the operation of “Standard Conditions I” at low potential, which effectively inhibited the competitive HER, and thus achieved the high chemoselectivity of imine hydrogenation. However, the reducing power of **9-HTX**^{-*} (**4**) is limited to -2.7 V vs. SCE. To further improve the reducing power of the catalytic system, we developed “Standard Conditions II” which removes TfOH and catalyzes the reductive functionalization reaction with **TX** alone. Here, a small point worth noting is that the precatalyst for the reductive functionalization is **TX**, and not **9-HTXTF** (**1**, **TX** + TfOH).

During the study, we tried to estimate the excited state potential of **TX**²⁻. However, due to its high activity, it is extremely unstable and cannot be characterized by ¹H NMR, UV-vis, fluorescence, etc. Therefore, we cannot evaluate its excited state potential in the same way as we evaluated **9-HTX**⁻ (**3**). Thus, we tested its reducing ability with some challenging substrates [**42S** (-2.9 V vs SCE) and **43S** (-3.4 V vs SCE)] and found that its reducing ability could reach a potency even comparable to that of Na⁰ or Li⁰.

The reviewer's comments are very useful. We realized that it was misleading to the reader because no clear and reasonable explanation of the problem is given in the manuscript. Once again, we thank the reviewer for making such an important suggestion and helping us to better refine this manuscript.

Since it is experimentally impossible to evaluate its excited state potential, we use theoretical calculations to study the electron transfer process. Indeed, the calculations suggest that the reduction of **43S** (-3.4 V vs SCE), which has a lower potential than **42S** (-2.9 V vs SCE), is exergonic in the presence of photoexcited **TX**²⁻, showing a ΔG of -19.3 kcal/mol. Therefore, photoexcited **TX**²⁻, which exists under basic conditions, might have the ability to reduce precursors of compounds **42** and **43** (i.e., **42S** and **43S**, -2.9 V and -3.4 V vs SCE).

We think that the π - π stacking assembly between catalyst and substrate indicated by the reviewer is very possible, as this would be more beneficial for electron transfer between them. But we cannot find solid proof to prove it.

Overall, we have added the following explanation to the revised manuscript:

“The exergonic reduction of **43S** (precursors of compounds **43**, - 3.4 V vs. SCE), which has a lower potential than **42S** (precursors of compounds **42**, - 2.9 V vs. SCE), is indeed supported by the computations (For details, please see Supplementary Information).”

It has also been added to the Supplementary Information as shown below:

One-electron reduction of 43S

The free energy for **43S** to be reduced by photoexcited catalyst **TX^{2-*}** is -19.3 kcal/mol:

Table S5. Absolute free energies (a.u.) of calculated structures

Species	Free Energies	Species	Free Energies
1	-1935.68921	TfOH	-962.21272
2	-1935.82526	TfO ⁻	-961.80160
3	-974.12421	43S (MeCN)	-1120.49969
3'	-1935.90830	43S^{*-} (MeCN)	-1120.54227
4	-974.05195	TX^{2-*} (MeCN)	-973.57155
5	-941.15789	TX^{*-} (MeCN)	-973.55970
5A	-1903.40389		
5B	-941.71123		
5C	-941.77453		
6	-942.34324		
TS-1	-1915.71694		

Coordinates of optimized structures

43S (solvent = MeCN)

C	1.19906900	1.45096400	-0.05180300
C	-0.02191400	2.08628000	-0.14163400
C	-1.20007000	1.38867900	-0.07037600
C	-1.19966200	-0.00265400	0.03360800
C	0.04414800	-0.64474400	-0.03088000
C	1.26001300	0.06826900	0.02038300
H	2.09654400	2.04665300	-0.04148000
H	-2.13147400	1.93238900	-0.09015000
C	-0.10503500	-2.51882500	-1.43515300
H	0.10980000	-3.58525000	-1.40128000
H	-1.13629100	-2.37238000	-1.75510400

H	0.55530500	-2.03885000	-2.15870300
C	-2.57185300	-0.67463700	0.24621300
C	-2.51923400	-2.12490800	0.73767200
H	-1.88364800	-2.22628200	1.61689000
H	-3.53078200	-2.42252100	1.01908700
H	-2.17294400	-2.82536700	-0.01477800
C	-3.38785400	-0.60746100	-1.04947000
H	-4.37416800	-1.04836500	-0.89125500
H	-3.52861700	0.42469600	-1.37298800
H	-2.90169200	-1.15067900	-1.85979900
C	-3.33190600	0.10049200	1.33765500
H	-2.76547600	0.11636100	2.27052800
H	-3.55158400	1.12807900	1.05564700
H	-4.28480600	-0.39422600	1.52964100
C	2.61527200	-0.62672800	0.23430800
C	2.99839300	-1.54761900	-0.93021800
H	4.03062900	-1.87791800	-0.79975700
H	2.37334300	-2.43342100	-0.97687300
H	2.93458500	-1.02408400	-1.88564400
C	2.55921200	-1.43975000	1.53499700
H	1.79216100	-2.20925700	1.49300200
H	3.52257800	-1.92380700	1.70771600
H	2.35269300	-0.78963200	2.38734800
C	3.74678900	0.39272700	0.38632000
H	3.89729200	0.97822600	-0.52192600
H	3.57238100	1.08069700	1.21434700
H	4.67386000	-0.14342500	0.59065800
O	0.11741100	-2.01236500	-0.12383400
Cl	-0.06477500	3.82686900	-0.29730600

43S⁻ (solvent = MeCN)

C	1.22537400	1.45605700	-0.27740200
C	0.03026500	2.09174300	-0.13995200
C	-1.18447700	1.44172000	0.10256500
C	-1.18832000	-0.00375700	0.10963800
C	0.01327000	-0.64475400	-0.07543000
C	1.29320200	0.02331800	-0.19693900
H	2.10985800	2.04492700	-0.47223100
H	-2.07864300	2.01513800	0.27332300
C	-0.05280000	-2.48424600	-1.53178200
H	0.12616600	-3.55981500	-1.53313600
H	-1.04808900	-2.29195100	-1.94176800
H	0.68436400	-2.00052400	-2.17739500
C	-2.57212700	-0.66618900	0.25449300
C	-2.56599500	-2.15628000	0.61364600
H	-1.95457900	-2.35328100	1.49405000
H	-3.59107200	-2.46031900	0.83948700
H	-2.20626800	-2.78844800	-0.19154100
C	-3.36855600	-0.48824400	-1.04578600
H	-4.37649700	-0.89717500	-0.93690000
H	-3.45910100	0.56614400	-1.31005300
H	-2.88466700	-1.00292400	-1.87725800
C	-3.35252900	0.02335900	1.38792400
H	-2.80478000	-0.03635200	2.33086800
H	-3.55836500	1.07177600	1.18366300
H	-4.31335500	-0.47650200	1.52569600
C	2.58968400	-0.66079500	0.26935600
C	3.04498000	-1.81809900	-0.63235200
H	4.00725100	-2.20733000	-0.28706100
H	2.33303900	-2.63846900	-0.62621100
H	3.17243000	-1.48126100	-1.66328700

C	2.40444000	-1.20744200	1.69465300
H	1.60645800	-1.94903100	1.72408700
H	3.32194500	-1.68016200	2.05813600
H	2.14610000	-0.40170200	2.38569400
C	3.75212000	0.33485200	0.31041200
H	3.95642500	0.75841100	-0.67520400
H	3.56127100	1.15946000	0.99804000
H	4.65774700	-0.17488200	0.64569800
O	0.05379300	-2.03369200	-0.19756700
Cl	-0.01465700	3.85919700	-0.30849600

TX²⁻* (solvent = MeCN)

S	0.00000000	-1.80940300	0.33780900
O	0.00000000	2.64389000	0.15021500
C	-1.36793600	-0.79346300	-0.01223400
C	-2.61413700	-1.39412200	-0.16675600
H	-2.66510200	-2.47803500	-0.21971800
C	-3.78548100	-0.65265900	-0.22885000
H	-4.73870900	-1.14913900	-0.36298800
C	-3.70593300	0.75998000	-0.11911000
H	-4.60751900	1.36103700	-0.14119400
C	-2.47468800	1.36753700	0.00530800
H	-2.40181200	2.44592100	0.07233500
C	-1.25769000	0.65071700	0.03430500
C	0.00000000	1.36397800	0.09074900
C	1.25769000	0.65071700	0.03430200
C	1.36793700	-0.79346300	-0.01223200
C	2.47468800	1.36753700	0.00530500
H	2.40181100	2.44592200	0.07232900
C	3.70593300	0.75998000	-0.11911200
H	4.60751900	1.36103800	-0.14119800

C	2.61413800	-1.39412200	-0.16675200
H	2.66510200	-2.47803600	-0.21971100
C	3.78548100	-0.65265900	-0.22884800
H	4.73871000	-1.14913800	-0.36298500

TX⁺ (solvent = MeCN)

S	1.82567700	0.34233800	0.00000000
O	-2.61874900	0.30899300	0.00000000
C	0.77219300	0.04155900	1.37801200
C	1.38365900	-0.14149800	2.60490900
H	2.46561400	-0.16940000	2.66092000
C	0.62798400	-0.27219400	3.76564800
H	1.11937100	-0.40321500	4.72057800
C	-0.75689100	-0.23137000	3.67447700
H	-1.36160000	-0.33030200	4.56786200
C	-1.37037900	-0.07111800	2.44792400
H	-2.44771200	-0.04608900	2.36589300
C	-0.63293000	0.06852000	1.25552900
C	-1.34930300	0.20364500	0.00000000
C	-0.63293000	0.06852000	-1.25552900
C	0.77219300	0.04155900	-1.37801200
C	-1.37037900	-0.07111800	-2.44792400
H	-2.44771200	-0.04608900	-2.36589300
C	-0.75689100	-0.23137000	-3.67447700
H	-1.36160000	-0.33030200	-4.56786200
C	1.38365900	-0.14149800	-2.60490900
H	2.46561400	-0.16940000	-2.66092000
C	0.62798400	-0.27219400	-3.76564800
H	1.11937100	-0.40321500	-4.72057800

Question 3:

- The authors are pushing a concept of “2e EPC”. However, this is conceptually misleading readers into thinking that the each catalytic cycle involves a two electron transformation of the catalyst. In fact, it is still a 1 electron EPC mechanism switching between an anion and a radical. This conceptual part of the introduction needs to be redrawn in a way that actually represents the discovery.

Response:

First of all, we thank the reviewer again for pointing out the errors for us. In the previous edition, Fig. 1c, which we presented to the reader, is indeed the single-electron transfer mode ($1e^-$ EPC). In fact, after careful modification, our mechanism has been adjusted to Fig 3g. However, we did not adjust Fig. 1c accordingly.

Original Fig. 1c:

After careful thought, we wished to explain to the reviewer. The previously reported strategy was to obtain the open-shell catalytically active species **Cat $^{\cdot-}$** by one-electron reduction at the cathode (Fig. 1b).

Fig. 1b:

Our strategy is to obtain the long-lived closed-shell catalyst **Cat $^{\cdot-}$** by two-electron reduction at the cathode (Fig. 1c in the revised manuscript).

Fig. 1c:

After adjustment, Fig. 1c now represents our findings and is consistent with Fig. 3g.

It has been revised in the manuscript as shown below:

Question 4:

- An analogous concept of consecutive PET is mentioned, and one review (Ref 22) is cited. However, this citation should come earlier upon the first mention (together with references 6,7). The existence of this concept is simply stated but not elaborated; no comparisons with the photoelectrochemistry – the main topic of this paper – are given (at least, the comparison is not clear to my eyes). A review should be added that compares the concepts: Beilstein J. Org. Chem. 2023, 19, 1055.

Response:

We have cited this review (previous reference 22) in the first mention of consecutive PET (present reference 8) as shown below:

“Consecutive photoinduced electron transfer (ConPET)⁶⁻⁸, which overcomes the energetic limitation of a single visible light photon, is another efficient and useful synthetic strategy and has been widely applied in

some high-energy demanding reactions like dehalogenation and further functionalization^{6,7,9-17}, pentafluorosulfanylation¹⁸, carboxylation^{19,20}, arene oxidation²¹, and Birch reduction²² under mild conditions.”

8. Glaser, F., Kerzig, C. & Wenger, O. S. Multi-Photon Excitation in Photoredox Catalysis: Concepts, Applications, Methods. *Angew. Chem. Int. Ed.* **59**, 10266-10284 (2020).

In addition, we have added a comparison of consecutive PET with photoelectrochemistry and cited the review (Beilstein J. Org. Chem. 2023, 19, 1055) as reference 62.

“Notably, although ConPET and PEC are different in the way of generating catalytically active species, they both have the same SET process for photoexcited active species and substrates⁶².

62. Lepori, M., Schmid, S. & Barham, J. P. Photoredox catalysis harvesting multiple photon or electrochemical energies. *Beilstein J. Org. Chem.* **19**, 1055-1145 (2023).

Question 5:

- Since authors discovered the reaction involved a photoexcited anion, a key review on photoexcited anions in organic transformations should really be cited: *Angew. Chem. Int. Ed.* 2020, 60, 6270. A recent paper from Melchiorre group (*Angew. Chem. Int. Ed.* 2023, 62, e202306364) that involves a photoexcited anion for unactivated aryl halide cleavages should be cited.

Response:

We are very grateful to the reviewer for the useful suggestions. we have cited the key review on photoexcited anions (*Angew. Chem. Int. Ed.* **2020**, 60, 6270) as reference 67. Furthermore, the paper from Melchiorre group (*Angew. Chem. Int. Ed.* **2023**, 62, e202306364) has been added to closed-shell photocatalyst as shown in reference 70.

It has been revised in the manuscript as shown below:

“In recent years, anionic or dianionic species have been gradually disclosed as closed-shell photocatalysts⁶⁷⁻⁷⁰.”

67. Schmalzbauer, M., Marcon, M. & König, B. Excited State Anions in Organic Transformations. *Angew. Chem. Int. Ed.* **60**, 6270-6292 (2021).

70. Wu, S., Schiel, F. & Melchiorre, P. A General Light-Driven Organocatalytic Platform for the Activation of Inert Substrates. *Angew. Chem. Int. Ed.* **62**, e202306364 (2023).

Question 6:

- The statement “Very recently, a few two-electron reduced closed-shell catalysts were disclosed” should be revisited/expanded upon to not mislead readers. Firstly, in both these reports, the reductant was prepared by chemical reduction (sodium bisulfite or NaBH₄), not electrochemical reduction. Although for Wenger’s report the phrase ‘two-electron reduction’ would be correct, a hydride reduction (Nocera report) is not the same as a two-electron reduction. Authors could instead mention ‘anionic or dianionic catalysts’. Secondly, the 9-TX anion herein achieved activation of even deactivated aryl halides (42 and 43), while Refs 66/67 were limited to activated aryl halides. Authors should mention that only systems proven to involve photoexcited radical anions could access unactivated aryl halides (ACS Catal. 2023, 13, 9392), and yet their system under review (9-TX anion) is a closed-shell photoreductant that can do so.

Response:

We are grateful to the reviewer for helping us avoid the incorrect description. We have changed “Very recently, a few two-electron reduced closed-shell catalysts were disclosed^{66,67}.” into “In recent years, anionic or dianionic species have been gradually disclosed as closed-shell photocatalysts⁶⁷⁻⁷⁰.”

In addition, we have mentioned that closed-shell photoreductant could access unactivated aryl halides that previously needed to be activated by photoexcited radical anions, as shown below:

“Notably, in previous reports, only systems proven to involve photoexcited radical anions could access unactivated aryl halides⁷⁸. Herein, such a closed-shell photoreductant could nicely achieve the above transformation.”

78. Horsewill, S. J., Hierlmeier, G., Farasat, Z., Barham, J. P. & Scott, D. J. Shining Fresh Light on Complex Photoredox Mechanisms through Isolation of Intermediate Radical Anions. *ACS Catal.* **13**, 9392-9403 (2023).

Question 7:

- In the pi-stacking complex TS-1, arene centroid distance should be shown. A distance of 4-6 Angstroms would support the notion of this interaction.

Response:

The calculations suggested that the ring-ring distance was 3.76 Å, which supported the significant interactions. The revised manuscript has presented this information as shown below:

**TS-1**

$$\Delta G^\ddagger = 11.3 \text{ kcal mol}^{-1}$$

Question 8:

- Authors should mention that a closed-shell catalyst with long lifetime that functions by two-electron cycling was recently reported for oxidative transformations: *Angew. Chem. Int. Ed.* 2023, e202307550. Exactly analogously, i) a benzophenone-type precatalyst is reduced to give the active photocatalyst and ii) presence of a strong acid caused a profound change in redox potential for cathodic reduction.

Response:

It has been revised as shown below:

“A long-lived closed-shell catalyst that functioned by two-electron cycling was also recently reported for oxidative transformations⁷¹.”

71. Žurauskas, J. *et al.* Electron-Poor Acridones and Acridiniums as Super Photooxidants in Molecular Photoelectrochemistry by Unusual Mechanisms. *Angew. Chem. Int. Ed.*, e202307550 (2023).

Question 9:

- (Regarding the response to another reviewer on the comparison of aryl halide reductions) I would encourage authors to highlight the beneficial advantages of their protocol within the manuscript itself, (undivided cell, Faradaic efficiency, no sacrificial reductants etc) as these could greatly benefit readers when

selecting reaction conditions. The ability to do these reactions in a simple undivided cell without sacrificial amine additives is indeed attractive.

Response:

We appreciate the reviewer for the good suggestions. We have highlighted the beneficial advantages of our protocol in the revised manuscript as shown below:

“In addition, compared with previously reported electrophotocatalytic coupling reactions^{51,52}, our protocol has the following advantages: (1) We achieved reductive functionalization of aryl halides in an undivided cell, avoiding the need for twice as many electrolytes and solvents to use divided cells; (2) We employed radical intermediates as sacrificial agents in an undivided cell, avoiding the use of additional terminal reductants; (3) Our catalytic system could significantly improve the faradaic efficiency of electrophotocatalytic reductive functionalization (for detail, please see Supplementary Information).”

51. Cowper, N. G. W., Chernowsky, C. P., Williams, O. P. & Wickens, Z. K. Potent Reductants via Electron-Primed Photoredox Catalysis: Unlocking Aryl Chlorides for Radical Coupling. *J. Am. Chem. Soc.* **142**, 2093-2099 (2020).

52. Kim, H., Kim, H., Lambert, T. H. & Lin, S. Reductive Electrophotocatalysis: Merging Electricity and Light to Achieve Extreme Reduction Potentials. *J. Am. Chem. Soc.* **142**, 2087-2092 (2020).

It has also been added to the Supplementary Information as shown below:

Compared with previously reported electrophotocatalytic coupling reactions^{6,8}

Using dicyanoanthracene or anthraquinone as electrophotocatalyst, Mg⁰ was sacrificed; Using naphthalene imides as electrophotocatalyst, Et₃N (2 equiv) was added to the anodic chamber as a terminal reductant; In our work, radical intermediates was used as sacrificial agents.

Using dicyanoanthracene as electrophotocatalyst, the faradaic efficiency was 35%; In our work, the faradaic efficiency was 66%.

Previous reports:

This work:

6. Cowper, N. G. W., Chernowsky, C. P., Williams, O. P. & Wickens, Z. K. Potent Reductants via Electron-Primed Photoredox Catalysis: Unlocking Aryl Chlorides for Radical Coupling. *J. Am. Chem. Soc.* **142**, 2093-2099 (2020).

8. Kim, H., Kim, H., Lambert, T. H. & Lin, S. Reductive Electrophotocatalysis: Merging Electricity and Light to Achieve Extreme Reduction Potentials. *J. Am. Chem. Soc.* **142**, 2087-2092 (2020).

Question 10:

- pg2, line 39. The authors should rephrase 'electron-primed' to 'electro-activated' as a more general term. Since, in many of the papers cited these are in an oxidative direction so the catalyst is not primed by an electron (rather, by a hole).

Response:

We fully agree with the reviewer, so we have changed “electron-primed” into “electro-activated” in the revised manuscript as shown below:

(1) “For example, C–H functionalization³⁹⁻⁵⁰, dehalogenation functionalization^{51,52}, alcohol oxidation⁵³, C–H diamination⁵⁴, olefin difunctionalization^{55,56}, reductive cleavage^{57,58}, C–F arylation⁵⁹, and enantioselective cyanation^{60,61} reactions were gradually developed *via* such an **electro-activated** photoredox catalysis strategy.”

(2) “Light on-off experiments indicated that light is critical for imine hydrogenation, supporting the **electro-activated** photoredox catalysis process (Fig. 3d, ground-state **3** generated *via* electrocatalysis cannot drive the transformation, but photoexcited **3** (i.e. **4**) can catalyze the reaction.)”

Question 11:

- Fig 5. The different colored circular/button symbols under the word ‘electrophotocatalysis’ all link to non-photoelectrochemical processes elsewhere in the scheme. Therefore it doesn’t make sense to me / is misleading to have these colored icons under the word ‘electrophotocatalysis’.

Response:

We have removed these colored icons under the word “electrophotocatalysis” in figure 5, as shown below:

Question 12:

- Since the emission of TX depends heavily on the reaction conditions (Figure 2g), the detection wavelength for the TCSPC analysis (Figure 3a) should be given as it is in Figure 2h.

Response:

It has been added to Figure 3a as shown below:

Reviewer #2:

The first revision of the work submitted for publication in Nature Communications entitled “Electrophotocatalysis: hydrogenation of imines and reductive functionalization of aryl halides” by Wen-Jie Kang, Yanbin Zhang, Bo Li and Hao Guo has answered in great detail all my demands and those of the other reviewers. Formatting errors aside, the discussion of how it affects temperature and/or light is simply very enriching. The triflate discussion is also appreciated. And the rest of discussion is also very fascinating. Thus, again I recommend publication.

Response:

We sincerely thank this reviewer for the recognition.

Reviewer #3:

The authors have made substantial corrections to the manuscript, removing some of the errors depicting intermediates. They have also provided evidence that neutral thioxanthenol itself is not active as a photoreductant. What seems clear is that the thioxanthenone is being reduced to a species that then reduces the imine substrates, and that the process is electrophotocatalytic.

I still find the invocation of a carbanion in the presence of more acidic bonds problematic. In particular, I don't see how structure **3** would exist under these, or really any, conditions. Why wouldn't the proton on the hydroxyl group migrate to the carbon? To put it another way, imagine taking thioxanthenol and treating it with 1 equiv of strong base. Would **3** form, or would the corresponding alkoxide? I think it's the latter. On the other hand, it seems possible that the oxygen is not protonated but rather coordinated to the aluminum ions that are invoked in the mechanism. Frankly, I don't know how to easily answer this question, but if the authors want to draw the carbanion as with structure **3**, I suppose they can do so.

In summary, I think that there is interesting reactivity here that is worth publishing. The mechanistic proposal is controversial in some respects, but seems like it must be accurate in its broad-stroke analysis. I support accepting this work.

Response:

CV, EPR, ^1H NMR, UV-vis, and calculations consistently supported that the reduced species of **1** was the closed-shell anionic **3** (Fig. 2). Electronic-structure calculations for **3** unveil that the highest occupied molecular orbital (HOMO) is delocalized over this 14-electron tricyclic aromatic system (Fig. 2f).

As shown in the Fig. 3g, during the reaction, **1** firstly obtains an electron from the cathode to form **2** which is a sp^2 hybridized carbon radical. Next, **2** gets another electron from the cathode. Notably, this electron must fill in the same orbital to form a large conjugated delocalized closed-shell anion species **3**. Even if another possible intermediate **3-1** could be generated, it would be formed by proton migration only after **3** was formed (Fig. A). Notably, **3** has a certain stability and is highly active. Under the reaction conditions, it will be quickly photoexcited, and then participates in the subsequent transformation and returns to **TX**. In contrast, the conversion of **3** to **3-1** by proton migration is a relatively slow process due to the disruption of the aromatic structure.

Fig. A

Stable 14-electron tricyclic aromatic system
Visible light absorption

Dearomatic system
Non-visible light absorption

In our reaction, we did add 10 mol % TfOH, but the pH value of the reaction solution was measured to be about 7. In such a neutral organic solvent system, Al^{3+} would be rapidly converted into $\text{Al}(\text{OH})_3$ in the presence of 3 equivalents of H_2O . Furthermore, we also observed that the aluminum was oxidized at the anode, and at the end of the reaction, a layer of $\text{Al}(\text{OH})_3$ precipitate attached to the aluminum surface. Therefore, we think that the coordination of aluminum ions with oxygen is less possible.

REVIEWERS' COMMENTS

Reviewer #1 (Remarks to the Author):

The revised manuscript of Kang, Zhang, Li and Guo perfectly addresses the previous concerns. Depiction of the mechanistic manifold is appropriate. It has been a pleasure to work with the authors on catching the issue with the aluminium and improving the clarity of the manuscript.

The manuscript should be published without further delay, I congratulate the authors on their outstanding work.